

# Millennial and orbital-scale variability in a 54,000-year record of total air content from the South Pole ice core

Jenna A. Epifanio[1], Edward J. Brook[1], Christo Buizert[1], Erin C. Pettit[1], Jon S. Edwards[1], John M. Fegyveresi[2], Todd A. Sowers[3], Jeffrey P. Severinghaus[4], Emma C. Kahle[5]

[1]College of Earth, Ocean, and Atmospheric Science, Oregon State University, Corvallis, 97331, USA
[2]School of Earth and Sustainability, Northern Arizona University, Flagstaff, 86011, USA
[3]The Earth and Environmental Systems Institute, Pennsylvania State University, University Park, PA, USA
[4]Scripps Institution of Oceanography, University of California San Diego, La Jolla, CA, USA
[5]Department of Earth and Space Sciences, University of Washington, Seattle, WA, USA

*Correspondence to*: Jenna A. Epifanio (jenna.epifanio@gmail.com)

**Abstract.** The total air content (TAC) of polar ice cores has long been considered a potential proxy for past ice sheet elevation. Recent work, however, has shown that a variety of other factors also influence this parameter. In this paper we present a high-resolution TAC record from the South Pole (SPC14) ice core covering the last 54,000 years and discuss the implications of the data for interpreting TAC from ice cores. The SPC14 TAC record shows 15   multiple features of interest, including (1) long-term orbital-scale variability, (2) millennial-scale variability in the Holocene and last glacial period, and (3) a period of stability from 35 ka to 25 ka. The longer, orbital-scale variations in TAC are highly correlated with integrated summer insolation (ISI), corroborating the potential of TAC to provide an independent dating tool via orbital tuning. Large millennial-scale variability in TAC during the last glacial period is positively correlated with past accumulation rate reconstructions as well as the $\delta^{15}N$ of 20   $N_2$, a firn thickness proxy. These TAC variations are too large to be controlled by direct effects of temperature and too rapid to be tied to elevation changes. We propose that grain size metamorphism near the firn surface is likely to explain these changes. We note, however, that at sites with different climate histories than the South Pole, TAC variations may be dominated by other processes. Our observations of millennial-scale variations in TAC show a different relationship with accumulation rate than observed at sites in Greenland.

## 25  1 Introduction

Total air content (TAC), the total quantity of air trapped in polar ice, has long been explored as a proxy for many past climate and paleo-environmental conditions. Air pressure, temperature, and pore volume at bubble close-off control TAC in polar ice cores, as described by the ideal gas law (Martinerie et al., 1992). Variations in



TAC can reflect changes in any of these parameters. Early work in polar ice cores focused on using TAC to

reconstruct ice sheet elevation changes given the dependence of atmospheric pressure on altitude (Raynaud and Lorius, 1973, Raynaud and Lebel, 1979). If TAC depended only on elevation it would provide an invaluable constraint on ice sheet models, informing our understanding of ice sheet dynamics and sea level change. However, while atmospheric pressure must influence TAC, the amount of air trapped in polar ice is also controlled by the pore volume at close-off, which has a complex relationship with local meteorological conditions as well as firn

densification, making TAC a particularly complicated proxy (Eicher et al., 2016, Gregory et al., 2014).

The pore volume at close-off is primarily controlled by firn densification processes and is not easily predicted. Previous studies have shown that variations in pore volume at bubble close-off appear to dwarf the influence of atmospheric surface pressure variations (Schwander et al., 1989, Eicher et al., 2016, Martiniere et al., 1994, Raynaud and Lebel, 1979, Krinner et al.,2000). For temperature, Martiniere et al. (1992) demonstrated a

spatial correlation between site temperature and pore volume at close-off, using data from late Holocene ice core samples. This effect of temperature on pore volume can be empirically predicted and counteracts the effect of the local temperature from the ideal gas law. The two temperature effects on TAC have been shown to nearly cancel each other (Raynaud et al., 2007).

Previous research also showed that TAC is significantly impacted by local solar insolation on orbital

time scales (Raynaud et al., 2007, Eicher et al., 2016). The proposed mechanism for this relationship requires that higher local summer insolation increases the size of snow grains in the first few meters of firn, which then decreases the pore volume in these same layers as they reach bubble close-off (Raynaud et al., 1997, Arnaud, 2000). Studies at low accumulation rate polar sites have confirmed that larger firn grain sizes decrease the porosity at bubble close-off (Gregory et al, 2014, Courville et al., 2007).  The relationship between TAC and local solar

insolation on orbital time scales raises the possibility of directly dating TAC records using orbital tuning (Raynaud et al., 1997, Bazin et al., 2013).

On shorter millennial timescales, in Greenland ice core records, Eicher et al. (2016) proposed that changes in overburden pressure of the overlying firn column, driven by changes in accumulation rate, drive an inverse relationship between pore volume at close-off and accumulation rate. This process is thought to increase

in importance with increasing temperature, rivalling the direct impact of changes in temperature. Eicher et al., (2016) suggest that the accumulation rate effect on TAC may be responsible for the small millennial-scale TAC variations observed in Greenland ice cores. The observation is at odds with research in the low-accumulation rate Megadunes region of Antarctica, which seems to indicate that accumulation rate and pore volume at close-off are positively correlated due to larger grain sizes in the firn at ultra-low accumulation rate sites (Gregory et al., 2014,



Courville et al., 2007). However, the near-zero accumulation rate at Megadunes allows for extraordinary firn metamorphism (over hundreds of years) with the consequent reduction of pore volume, a no-analogue situation for Greenland.

Because previous work shows that TAC varies on multiple time scales, high-resolution data are important for fully interpreting the TAC records. To this end, we made high-resolution measurements of TAC in the South

Pole ice core (SPC14). These measurements were made using a wet-extraction technique coupled with a vacuum line system that also provides the atmospheric methane ($CH_4$) concentration in the same ice core (Epifanio, et al., 2020).

The SPC14 ice core was drilled in 2014/15 over two field seasons and reached a depth of 1751 m. The SPC14 record spans the last 54,300 years (Winski et al., 2019, Epifanio et al., 2020). The data set provides a

valuable context for better constraining the complex controls on TAC in ice cores. The record shows long-term and short-term variations (Figure 1) which would not be resolved without our high-resolution measurements. In the discussion below, we compare the TAC record to solar insolation curves, as well as a variety of known climate proxies. We present a simple linear regression that explains variability in our record as a function of insolation and several other proxies. Our study supports the insolation control on TAC but also shows that other firn

densification-related processes create significant millennial-scale variability, underscoring the need for high-resolution data to use TAC records for constraints on gas trapping, ice core chronologies, and even past ice-sheet elevation changes.

## 2 Materials and Methods

### 2.1 Total air content measurements

The total air content (TAC) record from the SPC14 ice core was measured jointly in the Oregon State University (OSU) and Pennsylvania State University (PSU) ice core labs concurrently with measurements of $CH_4$. The core was discretely sampled at approximately 1-meter resolution. Methods followed previous work (Epifanio, et al., 2020, Mitchell et al., 2011, Lee et al., 2020, Buizert et al., 2021) with some calibration updates described

below. At OSU, the samples were measured in accordance with previously published methods (Mitchell et al., 2015, Buizert et al., 2021). The PSU samples were also measured concurrently with $CH_4$, using a wet extraction system. The differences in methods between OSU and PSU wet extraction systems are described in (Mischler et al., 2009, Fegyveresi, 2015, Epifanio et al., 2020, and WAIS Divide project members, 2013).



The OSU vacuum line is set up as portrayed in Figure 2. Samples are melted under vacuum in a two-

sided array of twelve glass vacuum flasks. The air released during the melt step is trapped in the headspace of the

flasks. The melted samples are then refrozen in a -70 °C ethanol bath. Once samples are refrozen, the air trapped

in the flask headspace is expanded through the array volume, isolation volume and into the gas chromatograph

(GC) sample loop. In the equations below, the quantity of air in the entire vacuum line (in moles) at this point is

designated $n_1$. The relevant array valve is then closed, trapping air upstream of the isolation and GC volumes

(Figure 2). This quantity of air is designated $n_2$. The pressure in the GC volume is recorded by a capacitance

manometer (rated at 0-100 torr, 1.5% accuracy, $P_1$) The air in the GC sample loop is then injected into the GC

column and measured for $CH_4$ concentration using a 6890N Agilent gas chromatograph with flame ionization

detector.  Once the analysis is complete, the gas remaining in the GC and isolation volumes (Figure 2) is evacuated.

The remaining air from the relevant array is then expanded into the GC section, and the remaining air in the array

volume is again isolated as described above. In total, we make four expansions and four measurements of $CH_4$

concentration, and four measurements of GC pressure per sample. We assume that the temperatures of the vacuum

line and GC do not change during the subsequent expansions. TAC is derived from the sample mass and pressure

measurements during the analysis, following the ideal gas law:

$$n = \frac{P_1 V_1}{R T_{eff}},$$    (1)

Where $P_1$ = Pressure, $V_1$ is volume, $n$ is moles of air, $T_{eff}$ is effective temperature (K) or weighted mean temperature

of the various sections of the vacuum line, and $R$ is the universal gas constant, 8.314 J/molK.

The effective temperature of the line can change from day to day due to differences in air temperature,

ethanol bath temperature, chiller efficiency, and flask headspace variation due to sample size. Because of the daily

variation, it is problematic to use a weighted average calculation of $T_{eff}$ in our calculation of TAC, as this would

require a daily estimate of $T_{eff}$ or an assumption that it is constant. To avoid this issue we take advantage of the

fact that we do multiple expansions on the vacuum line to make multiple $CH_4$ measurements, and then use the

ratio of expansion pressures to calibrate the TAC measurement, as follows.

Because the amount of air in the first expansion is equal to the amount in the second expansion plus what was

removed by evacuation:





$n_1 = n_2 + n_{rem}$ (2)

Where $n_1$ is the amount of air in the line during the first expansion, $n_2$ is the amount of air in the line during the second expansion, and $n_{rem}$ is the amount of air removed in the first expansion, and:

$n_1 = \frac{P_1 V_T}{R T_{eff}}, \; n_2 = \frac{P_2 V_T}{R T_{eff}}, \; n_{rem} = \frac{P_1 V_{gc}}{R T_{gc}}$ (3)

Where $P_1$ is the pressure of the first expansion, $V_T$ is the total volume of the vacuum line and GC, and $V_{gc}$ and $T_{gc}$ are the volume and temperature of the vacuum line past the GC valve. Following from equation (2),

$\frac{P_1 V_T}{R T_{eff}} - \frac{P_2 V_T}{R T_{eff}} = \frac{P_1 V_{gc}}{R T_{gc}}$ (4)

Rearranging to solve for $\frac{T_{gc}}{V_{gc}}$ :

$\frac{T_{gc}}{V_{gc}} = \frac{T_{eff} P_1}{V_T (P_1 - P_2)}$ (5)


and then rearranging to solve for $T_{eff}$

$T_{eff} = V_T \frac{P_1 - P_2}{P_1} \frac{T_{gc}}{V_{gc}}$ (6)

TAC is calculated by dividing $n_1$ by sample mass. To eliminate the need to measure $T_{eff}$ every time TAC is calculated, we use Teff from equation 6 in the calculation of TAC:

$\frac{n_1}{m} = \frac{P_1^2}{R(P_1 - P_2)} \frac{T_{gc}}{m V_{gc}}$ (7)

We then substitute R in equation (7) for the ideal gas law, with a final calculation for TAC:



$$TAC = \frac{P_1^2}{R(P_1-P_2)} \frac{T_{gc}T_{STP}}{mV_{gc}P_{STP}} \qquad (8)$$

Equation (8) requires that we have an accurate measure of $\frac{T_{gc}}{V_{gc}}$. Two separate methods were used to

achieve this. The first method involved estimating $T_{eff}$ by taking a weighted average of the system temperature with the GC oven at 50°C and the flasks submerged in a -70°C ethanol bath. Once a known amount of air was expanded into the GC, $\frac{T_{gc}}{V_{gc}}$ was calculated. This relied on an accurate estimation of the line temperature for the portion outside of the GC oven. The value of $\frac{T_{gc}}{V_{gc}}$ determined using this method was is 12.37 (K/cc).

The second calibration did not use the chilled flasks and instead utilized a removable known volume of air that was externally attached to the vacuum line. From expansions of air to other parts of the array and GC, we were able to calculate the volume of the entire system and individual components and the GC. All volumes are listed in Table 1. Air was expanded into the entire system at a known pressure, with the GC at room temperature (21.5°C). The front array valve was closed, isolating the pressure in the front array, while the rest of the line was

pumped to vacuum. After the pressure stabilized, the air in the front array was expanded to the entire line, and the pressure was recorded. Finally, the GC oven was switched on and raised to the normal operating temperature (50°C). The pressure in the total line after heating was recorded as $P_{eff}$. To solve for $T_{eff}$, we use the combined gas law:

$$\frac{P_1V_1}{T_1} = \frac{P_2V_2}{T_2} \qquad (9)$$

Where $P_1 = P_{array}$ is the pressure of the air in the front array with the line at room temperature, $V_1 = V_{array}$ is the volume of the front array, $T_1$ is the measured lab temperature, $P_2 = P_{eff}$ or the pressure of the air in the entire system, including both array volumes, isolation volume, and the GC volume, and $T_2 = T_{eff}$ or the effective

temperature of the entire system with the GC at 50°C. Rearranging equation (9) to solve for $T_2$ (or $T_{eff}$):

$$T_{eff} = \frac{P_{eff}V_T T_{array}}{P_{array}V_{array}} \qquad (10)$$



Finally, multiple expansions of air from the front array to the GC volume give a consistent ratio of

pressures of 0.56, which combined with equation (5), yields a final $\frac{T_{gc}}{V_{gc}}$ of 12.79 (K/cc). This value is different

from the original value of 12.37 (K/cc) but we believe is more accurate due to the difficulty of estimating $T_{eff}$

during the first calibration. The second method also removes the variability of flask size and accounts for small

changes made in the vacuum line over time. All measurements of TAC on the SPC14 ice core were calibrated

using the $\frac{T_{gc}}{V_{gc}}$ value of 12.79 (K/cc).


### 2.2 Solubility and cut bubble corrections

Two small corrections were made to the TAC measurements to account for effects on TAC introduced during

sample preparation and measurement. Because the air trapped in the headspace of the flask is not allowed

sufficient time to reach solubility equilibrium during the melt refreeze process, a correction was made to account

for the residual gas trapped in the refrozen ice. Following Mitchell et al. (2015), we determined an empirical TAC

solubility correction of +1.3%, which we added to all sample measurements. This empirical correction was derived

during wet extraction by measuring the TAC remaining within the flask headspace after a second melt-refreeze

cycle.

The second correction we applied to our TAC data was to account for the air lost during cutting and

trimming the ice samples. Preparing ice-core samples for measurement unavoidably cuts through some bubbles,

releasing a fraction of the air content from the entire ice sample. As a result, measurements made of the total air

are slightly lower than true values.

This cut-bubble correction is based on a statistical relationship among the total number and average size

of bubbles in a given sample, and the amount of exposed surface area that is cut during sample preparation

(Saltykov, 1976). Bubble numbers and average sizes were determined during number-density and physical

property measurements as described in Fegyveresi et al. (2011) and Fitzpatrick et al., (2014). Additional three-

dimensional micro-CT measurements were made of selected samples to validate bubble numbers and sizes

(Fegyveresi et al., in prep). Similar to the methods in Matinerie et al., (1990) and Fegyveresi (2015), we averaged

the cut-bubble correction for each sample and applied it across our dataset. We calculated a cut-bubble correction

that gives a maximum of 8% loss in the first 200 m of ice, decreasing to 1.9% TAC loss at the base of the bubbly

ice at ~1200m (at the onset of the clathrate-ice transition). We applied no correction after the base of the bubbly

ice.





**3 Results and Discussion**

**3.1 South Pole total air content record**

The SPC14 ice core TAC record is shown in Figure 1, plotted on the SPC19 ice age scale (Winski et al., 2019). We plot our data on the ice-age scale, rather than the gas-age scale, given evidence that ice-sheet surface conditions have a major impact on the pore volume at bubble close-off, further discussed below. This convention is consistent with previous research (Eicher et al, 2016, Raynaud et al, 2007). Using the method described above, we measured TAC concurrently with the SPC14 $CH_4$ (Epifanio et al., 2020), resulting in a total of 2,318 measurements made on samples at 1,067 individual depths. Samples were taken at approximately every meter along the depth range. The depth range is from 131 m to 1,751 m, which spans the period of 130 to 54,302 years BP. The age resolution of these samples is the same as reported for $CH_4$ in the SPC14 ice core, averaging 42 years, but increasing from 20 years in the Holocene to 190 years at the bottom of the ice core record. Differences in methods between the OSU and PSU labs created a mean offset of 0.0072 $cm^3/g$. To correct for this offset, PSU values were increased to be comparable to OSU values. The pooled standard deviation, after correction, is 0.002 $cm^3/g$, which is approximately 2% of the mean value. Data are available at the USAP data repository (Epifanio, et al., 2022). To our knowledge, this is the first ice core TAC record with this resolution and length, allowing in-depth comparison with other climate proxies at a site that is not likely to have experienced significant elevation change over the last 54 ka (Fudge et al., 2020, Lilien et al., 2018).

The SPC14 TAC record has several notable characteristics (Figure 1). Long term trends include an increase in TAC from 53 ka to 36 ka, followed by an approximately 10 ka period of low variability, then decreasing TAC from about 20 ka to the present. Millennial-scale variations, both in the Holocene and glacial period, consist of large magnitude abrupt changes. These variations are as large as 0.007 $cm^3/g$ in 2,600 years, which is 47% of the total amplitude in the record. Higher frequency variations exist on centennial or shorter timescales that may represent firn processes such as layering, but our sampling resolution is not adequate to fully evaluate this possibility.

**3.2 Impacts of temperature on TAC**

Prior work suggests that temperature impacts TAC in two ways, the first through the ideal gas law, and the second through an empirically established relationship between the pore volume at bubble close-off and temperature (Martinerie et al., 1992). Given a fixed pore volume at close-off, TAC is given by the ideal gas law:



$$TAC = V_c * \frac{P_c}{T_c} * \frac{T_0}{P_0} \qquad (11)$$

Where TAC is the total air content (cm³/g), $V_c$ is the pore volume at close off (cm³/g), $P_c$ and $T_c$ are pressure and

temperature of the air contained in $V_c$ at close-off (mb and K), and $P_0$ and $T_0$ are standard pressure and temperature

(1013 mb and 273K, respectively).

The empirical relationship between pore volume and temperature, as found by Martinerie et al. (1992) is

given by:


$$V_c = 7.6x10^{-4} * T_s - 0.057 \qquad (12)$$

Where $T_s$ is the snow temperature (K).

To examine direct thermal effects, we follow Raynaud et al. (2007) and Eicher et al. (2016) to define a

non-thermal residual term $V_{cr}$:

$$V_{cr} = TAC * \frac{T_c}{P_c} * \frac{P_0}{T_0} - V_c \qquad (13)$$

For the temperature at bubble close-off ($T_s$), a temperature reconstruction from Kahle et al (2021). To

approximate $P_c$, we use value of 680 mb at the surface of the ice core, linearly decreasing with depth to account

for 180 m of elevation increase upstream of the drill site (NOAA GML, 2020, Fudge et al., 2020).

Following Raynaud et al. (2007) and Lipenkov et al. (2011), we then create standardized versions of

TAC and Vcr, TAC* and Vcr*. These data are plotted together in Figure 3. The correlation between TAC* and

Vcr* is very high ($r^2 = 0.96$).  Because Vcr* is a quantity that essentially describes TAC* in the absence of

temperature effects, Vcr* is a useful quantity to understand the magnitude of the direct effects of temperature. The

very-high correlation between TAC* and Vcr* agrees with previous work on both EDC and NGRIP ice cores that

show the direct temperature effect on TAC is responsible for a very small amount of TAC variability. As a result,

we targeted our analysis using measured TAC instead of the non-thermal residual.

### 3.3 Orbital-scale variations in TAC

Local integrated summer insolation (ISI) has shown an anti-correlation with TAC in various Antarctic

and Greenland ice cores (Raynaud and Lebel, 1979, Raynaud et al., 1997, Eicher et al., 2016). South Pole is an



interesting site to study in this regard, as it is the only place in Antarctica without a diurnal cycle in solar insolation. Following Huybers et al., (2006) and Raynaud et al., (2007) we find ISI by:

$$ISI = \sum \beta_i(\omega_i * 86,400) \qquad (14)$$

where ISI is in joules, $\omega_i$ is the daily insolation in W/m², and $\beta_i$ is a Heaviside step function, where $\beta_i = 1$ when $\omega_i$ is $> \omega_{threshold}$, otherwise $\beta_i = 0$. ISI is characterized by a mix of precession and obliquity and are unique for latitude and the chosen $\omega_{threshold}$.

Earlier research discusses the anti-correlation between ISI and TAC and uses the relationship to orbitally tune the ages of Antarctic ice core records (Lipenkov et al., 2011, Raynaud et al., 2007, Parrenin et al., 2007). Conclusions of previous work suggest that the insolation effect is much larger than the direct effects of temperature discussed above and is found in both Northern and Southern Hemisphere records (Eicher et al., 2016).

We find a maximum anti-correlation of ISI with TAC at a cutoff threshold of 225 W/m² ($r^2 = 0.46$, p < 275   0.0001). Studies of the same relationship in the EDC and NGRIP ice cores found maximum correlations at different cutoff values (390 W/m² for NGRIP with $r^2 = 0.3$) (Eicher et al., 2016, Raynaud et al., 2007). However, the correlation difference between varying threshold values (0 to 500 W/m²) is small ($r^2 = 0.42$ to $r^2 = 0.46$). The linear regression of TAC as a function of ISI at a threshold cutoff of 225 W/m² is shown in Figure 4. To make this calculation, we interpolated the ISI data to the ages assigned to TAC.

The ISI regression misfit values show an even distribution. The link between TAC and ISI supports the proposition that TAC can be used for absolute (orbitally tuned) dating of ice cores (Raynaud et al., 2007). It is worth noting, however, that the SPC14 TAC record only covers the last 54,000 years, which is only slightly longer than one obliquity cycle. This time period is shorter than the ice core records mentioned above, which may affect the differences in cutoff threshold that give maximum correlation between TAC and ISI.


### 3.4 Millennial-scale variations observed in TAC

While the ISI appears to be highly correlated with long-term variations in TAC, it fails to explain the observed millennial-scale changes. Abrupt, large magnitude, millennial-scale variations in TAC exist through the Holocene and early part of the glacial period. The approximate magnitude of the largest, abrupt. millennial-scale 290   changes is 0.007 cm³/g, which is similar to the abrupt millennial-scale variations observed in NGRIP, which were typically around 0.01 cm³/g (Eicher et al., 2016). Millennial-scale variations are absent from 40 ka to 22 ka in the



SPC14 ice core, but high frequency (centennial-scale and smaller) variations persist through the entire record. The existence of millennial-scale and higher frequency variations underscores the need for high-resolution data to discern the long-term patterns from other climatic effects on TAC. We isolate the millennial-scale TAC variations

by subtracting the orbital-scale trends that we find via regression of TAC onto ISI (Fig. 5). The largest millennial-scale features occur at 52 ka, 46 ka, and 44 ka. Similar large features are not found between 43 ka and 25 ka, where the TAC reaches a maximum, steady value of about 0.11 cm³/g. Large variations resume at 22 ka and occur through the Holocene time period, mostly between 9 ka and 2 ka.

We first investigate the possibility of surface pressure changes driving the millennial-scale variability.

Using the relationship of air pressure with TAC (Eq. 11) as proposed by Martinerie et al., (1992), we calculated the pressure change required for a representative high-frequency TAC change. For example, the abrupt increase in TAC of 0.01 cm³/g in 2.7 kyr starting at 49.2 ka (Figure 1) would require a mean pressure change of 104 hPa over that time interval. In the absence of elevation change, central Eastern Antarctica could not likely sustain large atmospheric pressure changes for that length of time. Hourly atmospheric pressure measurements at the South

Pole are between 710 and 650 hPa, varying on seasonal timescales (NOAA GML, 2020). However, daily, or even seasonal, variations in pressure are not likely to be recorded in our TAC record due to the gradual bubble trapping process that takes decades to complete. Current spatial gradients of sea level pressure, including variations associated with the Amundsen Sea Low (ASL), show a maximum range of about 50 mb from max low to high pressure over Antarctica (Schneider et al., 2012, Raphael et al., 2016). To create a change in atmospheric pressure

large enough to explain these TAC variations would require a dramatic reorganization of the westerly winds, which is highly unlikely during the last glacial period. Some modeling studies have examined wind changes around Antarctica, using 10,000-year model runs, and found moderate changes in sea level pressure, no greater than 15 hPa (Goodwin et al., 2014).

If the TAC change were entirely explained by a surface pressure change induced from ice sheet elevation

changes, an elevation change of about 1,200 meters would be needed. Such a large and abrupt change is unlikely to have been driven by accumulation change alone and is otherwise not likely due to the geographical context of the South Pole. Accumulation rates during the glacial period were just 3 cm/yr (water equivalent), which would mean the maximum amount of accumulation gain for our sample abrupt change (2.64 kyr) would be only about 80 meters, without considering the effects of ice layer thinning.

The location of SPC14 in the interior of Eastern Antarctica means that it would not have been subject to large elevation changes associated with major climate transitions, especially during the beginning of the last glacial period (Golledge et al., 2014, Fudge et al., 2020). The SPC14 ice core was not drilled at an ice divide,

 

which means the ice collected at the South Pole flowed over topography before it arrived at the sampling location. This flow changed the local elevation at which the snow fell in the past (the bedrock elevation history along the upstream flow line as it corresponds to the age of ice in the ice core is shown in Figure 6). The history of ice flow of the SPC14 ice core, however, is not constrained beyond 100 km from the core site and is best constrained in the closest 65 km, representing the last 10 ka, or about 725 m ice core depth (Lilien et al., 2018, Fudge et al., 2020). Following the ice flow line upstream from the modern ice core site shows the bedrock elevation only varying by about 500 m, which corresponds to an ice surface elevation variation of about 300 meters during that time (Lilien et al., 2018, Fudge et al., 2020, Kahle et al., 2021, Morgan et al., 2022). This 300 m change is much less than the 1200 m change required by our millennial-scale TAC observations. Large transient features in surface ice sheet elevation do not exist, eliminating ice flow as a source of abrupt elevation change influencing TAC. Figure 6 shows the inferred bedrock elevation under the ice core location over time and its relationship to TAC.

Large transient features in surface ice sheet elevation do not exist, eliminating ice flow as a source of abrupt elevation change influencing TAC. Figure 6 shows the inferred bedrock elevation under the ice core location over time and its relationship to TAC. Another hypothesis for changing TAC is related to horizontal stresses in the ice is due to stretching of the firn, decreasing the close-off depth. However, we do not observe any obvious relationships in bedrock elevation changes that would support this idea (Figure 6). Finally, layering due to melt or other layering effects also influences the trapping of air in ice, influencing TAC. However, due to the lack of melt layers at this location, this possibility is beyond the scope of this study to investigate.

### 3.4.1 Accumulation

Since we can reasonably show that large millennial-scale changes in TAC are not due to direct effects of temperature, variations in air pressure or ice dynamics, pore volume at bubble close-off must account for most of the TAC variations. Two observations are important to help explain these changes as a function of pore volume at close-off: (1) a positive correlation between TAC and accumulation rate, and (2), a high positive correlation between TAC and $\delta^{15}N$-$N_2$.

First, the millennial-scale changes in TAC are highly correlated with the accumulation rate reconstruction from Kahle et al. (2021), also plotted on the SP19 ice age ($r^2 = 0.59$, $p < 0.001$, Figure 5 and Table 4). Kahle et al. (2021) calculated the accumulation rate by using an inverse method which used models of firn densification, water isotope diffusion, and layer thinning to predict an accumulation rate history based on constraints provided by measured $\Delta$age, diffusion length, and annual layer thicknesses. The first 11.3 ka of the Kahle et al. (2021) reconstruction incorporates measured annual layer thicknesses from the layer counted time scale from Winski et





al., (2019). This results in an accumulation record that closely resembles the previous accumulation rate history from Winski et al. for this section of the core. Note that Kahle et al., (2021) do not use the $\delta^{15}$N-N$_2$ data in their

inversion method.

Second, the millennial-scale changes in TAC, plotted on the SP19 ice age, are also highly correlated with $\delta^{15}$N-N$_2$ at all depths ($r^2 = 0.51$, p < 0.001, Figure 5 and Table 4).  This observation is consistent with the idea that millennial-scale fluctuations in accumulation drive similar scale changes dominated by changes in the gravitational fractionation, which in turn responds to changes in firn thickness (Severinghaus et al., 1998, Sowers

et al., 1992, Morgan et al., 2022). As temperature variations are relatively minor, accumulation variation drives the observed changes in SPC14 $\delta^{15}$N-N$_2$ with greater accumulation rates causing a thicker firn column and a higher $\delta^{15}$N-N$_2$.. Winski et al., (2019) notes the close resemblance of $\delta^{15}$N-N$_2$ and the Holocene accumulation rate reconstruction, which is further evidence to support the use of $\delta^{15}$N-N$_2$ when as an indicator of accumulation rate changes in SPC14.

Metamorphism of the ice in the first few meters of the firn may explain the link between accumulation rate and TAC. Lower accumulation rates allow grains to remain at or near the surface for a longer time, giving grains in the firn more time to grow while they remain at the surface (Courville et al., 2007). The size of the firn grains at the surface seems to predict at which density bubble close-off occurs, with larger grained firn closing off at a higher density (Gregory et al, 2014). Because ice density is, by definition, inversely proportional to porosity,

higher density bubble close-off (associated here with larger grain sizes and lower accumulation) leads to bubbles with less pore volume than firn with smaller grain sizes. Additionally, lower accumulation rates may allow more time for grains to become spherically shaped, where higher accumulation rates tend to close off bubbles earlier in the densification process. Low accumulation rates create more homogeneous, spherically shaped grains which force more air to escape the ice core, leading to lower TAC. We propose that a mechanism of grain size and shape

affecting pore volume leads to a positive correlation between accumulation and TAC, which we observe in the SPC14 ice core.

In a sense this proposed grain size mechanism is similar to the proposed mechanism for how ISI impacts TAC (Raynaud et al., 2007, Eicher et al., 2016). ISI is hypothesized to act on TAC by changing the grain size of the firn at the surface by influencing temperature gradients in the first few meters of firn. Higher ISI increases the

near-surface firn metamorphism and grain size, and decreases pore volume at close off, resulting in the inverse relationship between TAC and ISI recorded in both hemispheres (Raynaud et al., 1997, Eicher et al., 2016). In our proposed mechanism, lower accumulation increases near-surface firn metamorphism and grain size and decreases pore volume at close-off. In both scenarios, grain size is set in the first few meters of the firn and the impact is



advected to the close-off depth. We propose that the relationship between grain size and accumulation rate is
responsible for the large, millennial-scale changes in TAC found in the SPC14 ice core.

Studies of TAC in Greenland suggest a different mechanism for similar-magnitude changes in TAC.
Eicher et al. (2016) observed a complex, asynchronous relationship between accumulation rate and millennial-
scale TAC changes in the NGRIP ice core. They suggest the relationship is due to step changes in accumulation
rate and temperature during D-O events in Greenland, causing the firn to densify rapidly due to increased load
changes. This hypothesis is not consistent with our results in the SPC14 ice core, perhaps because the load changes
in Antarctica are not nearly as large or abrupt as observed in Greenland. Accumulation rate changes at the South
Pole during the glacial period are much smaller than in Greenland (1 cm/yr or 25% change over an AIM event,
compared with 10 cm/yr or 100% increase during a D-O event in Greenland), while the TAC changes at both
locations are of a similar magnitude (0.007 cm$^3$/g at SPC14, 0.005 cm$^3$/g at NGRIP). This suggests that the
accumulation effect on TAC scales differently at NGRIP when compared to SPC14, implying that the two
locations respond differently to accumulation changes. Interpreting TAC at any ice core requires a universal
explanation of how the firn densifies, and future work may elucidate a single mechanism that can explain the
TAC-accumulation relationship at both polar regions. For example, we hypothesize that the grain size mechanism
dominates at colder, dryer locations, and the mechanism requiring transient firn densification would prevail when
warmer, wetter conditions exist. High resolution TAC datasets across a suite of climate conditions are needed to
test this hypothesis.

**3.5 Multiple regression to further investigate controls on TAC**

A multiple-regression analysis was performed to examine how climate-related variables correlate with
TAC to examine the possibility of removing non-elevation dependent signals from a future TAC record. Because
we do not expect large elevation changes at the South Pole site, SPC14 is an excellent ice core to examine this
possibility. If the TAC variability in the SP14 core can be explained using measured or modeled climate variables,
it might be possible in future projects to extract the portion of the variability due to elevation change. Here,
variables that might be related to TAC, either through changes in close-off volume or direct effects of the ideal
gas law, were considered in two separate multiple linear regression analyses. In the first multiple regression
(referred to as the 'modeled reconstruction' regression), we considered ISI, accumulation rate, temperature, and
the gas age – ice age difference or delta age (Δage). In the second multiple regression (referred to as the 'measured
data' multiple regression), we considered ISI, Δage, $\delta^{15}$N-N$_2$, and $\delta^{18}$O$_{ice}$. TAC data and variables considered are
plotted in Figure 7. Δage was employed because it represents the amount of time the firn is open to atmospheric




mixing; it is empirically determined in SPC14 from independent gas and ice age scales (Epifanio, et al., 2020),
removing modeling complications. All variables considered were either left in the regression or removed based
on the significance of fit as determined by a p value.

The modeled reconstruction regression included the Kahle et al. (2021) reconstructions of temperature
and accumulation rate, and ISI. The multiple regression had a maximum adjusted $r^2 = 0.72$ (p < 0.0001), therefore
the combined relationship accounts for 72% of the variation in the SPC14 TAC. The modeled reconstruction
multiple regression residuals show an even distribution. The parameters are listed in Table 2 in order of how much
each parameter affected the adjusted correlation coefficient. In the modeled reconstruction multiple regression,
ISI and accumulation accounted for 14% and 15% of the multiple regression, respectively. For example, removing
the ISI from the multiple-regression analysis, but leaving all other parameters in the analysis produces an $r^2$ of
0.58.

A regression using only measured parameters incorporated $\delta^{15}N\text{-}N_2$ and $\delta^{18}O_{ice}$ instead of using the
modeled accumulation rate and temperature reconstructions. Results for the measured data multiple regression are
listed in Table 1.3, again in the order of how much each variable changes the final multiple regression's adjusted
correlation coefficient. We find a maximum adjusted $r^2 = 0.69$ (p < 0.0001). In this case, ISI and $\delta^{15}N\text{-}N_2$ affect
the correlation coefficient the most, accounting for 10% and 8% of the adjusted correlation coefficient each. Both
the modeled and measured parameter multiple regressions compare well (Figure 8).

An interesting feature of this analysis is that if the derivative of Δage ($\delta\Delta age/\delta t$) is added to the multiple
regression, it seems to explain more of the variability observed in the TAC record. A comparison between a
regression that includes $\delta\Delta age/\delta t$, and a regression that does not include $\delta\Delta age/\delta t$ is shown in Figure 8.
Specifically, $\delta\Delta age/\delta t$ seems to correlate well with the magnitude of TAC change that occurs at 2,600 years as
well as the large variations that occur between 45 ka and the oldest part of the record. A regression analysis that
includes $\partial\Delta age/\partial t$ and the measured parameters (ISI, $\delta^{15}N\text{-}N_2$, $\delta^{18}O_{ice}$, and Δage ) gives an adjusted $r^2$ of 0.77 (p <
0.0001), meaning that $\delta\Delta age/\delta t$ and its interactions describe about 8% of the measured data multiple regression
solution. Adding $\delta\Delta age/\delta t$ to the modeled reconstruction multiple regression increases the $r^2$ adjusted by 4%.
While the reason $\delta\Delta age/\delta t$ helps explain TAC changes in the firn is not at first clear, times when $\delta\Delta age/\delta t$ are
large occurs at the same time as large changes in temperature and accumulation rate, specifically at D-O 12 and
13. This points to a mechanism in the firn column that responds to transient accumulation changes. Following the
reasoning of Eicher et al (2016), times of large changes in accumulation may not allow the firn to form spherical
bubbles, creating less space, and therefore lower TAC values.



An observation that is not explained by the multiple regression is the apparent lack of TAC variability
from around 25 ka to 42 ka. This section of TAC still responds to the influence of ISI, but seems to stop responding
to any other millennial-scale forcing. In fact, our multiple regression predicts millennial-scale changes during this
time that are not realized in the TAC record, most notably a large, predicted change around 32 ka that is not
reflected in TAC.

## 4 Conclusion

We present a high-resolution total air content record from the SPC14 ice core covering the last 54 kyr.
The implications of the analysis of the SPC14 TAC record are twofold. First, this study strongly confirms previous
research on Greenland (NGRIP and GRIP) ice cores as well as Antarctic ice cores (EDC) that low-frequency
variations in TAC depend on local summer insolation. Importantly, the high correlation with integrated summer
insolation (ISI) corroborates the potential to orbitally tune ice age scales using TAC. Second, because the SPC14
ice core was drilled at a location with relatively small expected elevation changes over the course of the ice core
history, it provides a good location to study controls on TAC outside the influence of elevation changes. Our
results suggest that temperature, accumulation rate, and other firn column properties interact together to affect
TAC and provide important context for developing paleoelevation proxies in cores that may have experienced
large elevation changes.

We propose that a common mechanism, grain size metamorphism in the top few meters of the firn, can
explain both orbital and millennial-scale times scale of TAC variations in the SPC14 ice core. Although our data
are consistent with an orbital scale impact of ISI on TAC, and an accumulation-driven millennial component,
further work is needed. Future directions should include:

(1) High-resolution sampling of TAC in multiple ice cores. High-resolution data are required to resolve
all millennial-scale features in TAC. These features need to be accounted for before understanding the elevation
history of a location. Data should be collected for both Northern and Southern Hemisphere ice cores, and at a
variety of accumulation rate regimes.

(2) Further understanding of links between $\delta^{15}$N-N$_2$, $\Delta$age, and TAC in ice cores. High-resolution
sampling of $\delta^{15}$N-N$_2$ in ice cores, and model-independent $\Delta$age determinations, could be used in future work to
corroborate findings from the SPC14 ice core. Additionally, TAC sampling at locations with well-known
accumulation rate histories will provide further constraints.



(3) Comparison of TAC in regions that have stable elevation histories to regions with unstable elevation histories. Ice cores targeted in coastal regions with hypothesized unstable elevation histories, such as in Western Antarctica, should be targeted for future TAC research.


**Data Availability**

The SPC14 total air content record is published on and can be accessed through the USAP Data Center with DOI: https://doi.org/10.15784/601546.


**Author Contributions**

JAE, EJB, JSE, TAS, JPS, JMF, ECK contributed data to this study. JAE, EJB, CB, JSE, TAS, measured ice core gases. ECK and JPS made isotope measurements. JMF made bubble density and size measurements. EJB and
TAS provided funding acquisition for the study. JAE, EJB, ECP, CB, JSE, JMF, JPS, wrote, reviewed and edited the paper with input from all authors.

**Competing Interests**

The authors declare that they have no conflict of interest.

**Acknowledgements**

This work has been funded by the National Science Foundation (Awards 1643722, 1443472, and 1443464). We
would like to thank Mark Twickler and Joe Souney for their work administering the SPICEcore project; the U.S. Ice Drilling Program for collecting the SPC14; the field team who collected the ice core; the National Ice Core Facility for ice core storage and processing; Michael Kalk for his help on lab work with the samples, and the many student researchers who produced data from the SPC14 ice core.






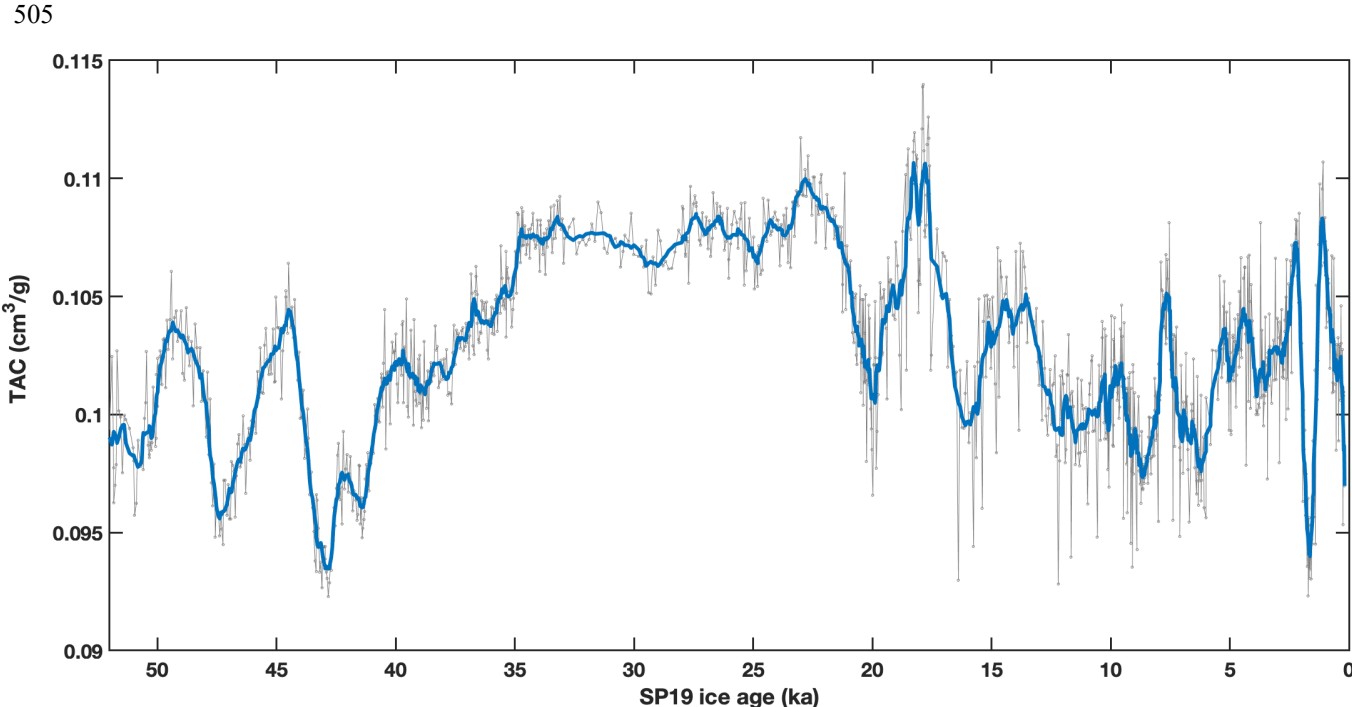

**Figure 1: Total air content of the SPC14 ice core.** Measurements are averaged duplicate measurements, plotted on the SP19 ice age scale (Winski et al., 2019). Blue line is smoothed record using a running 10-point average. TAC is expressed in units of cm³ air at standard temperature and pressure, per gram of ice.




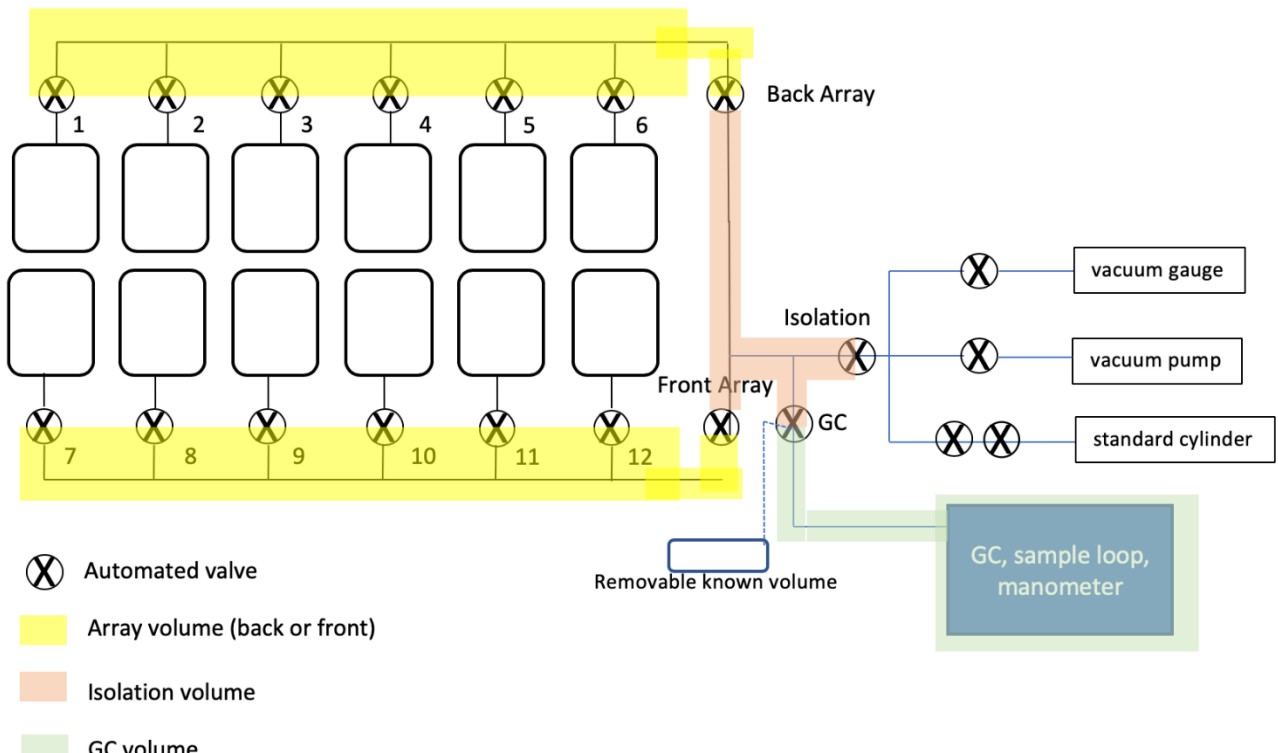

**Figure 2: Schematic of extraction line and GC for TAC and CH$_4$ measurements.** Diagram includes a removable known
volume, which was used in the $T_{eff}$ calibration (see text). Numbered valves are individual flask valves. Air released during the
melt phase is trapped between the flask and the numbered automated valve. Yellow: Back and front array volumes. Light
Orange: Isolation volume, includes the vacuum line between the array valves, the isolation valve, and the GC valve. Green:
GC volume, includes the vacuum line between the GC valve and the GC sample loop. All volumes are listed in Table 1.





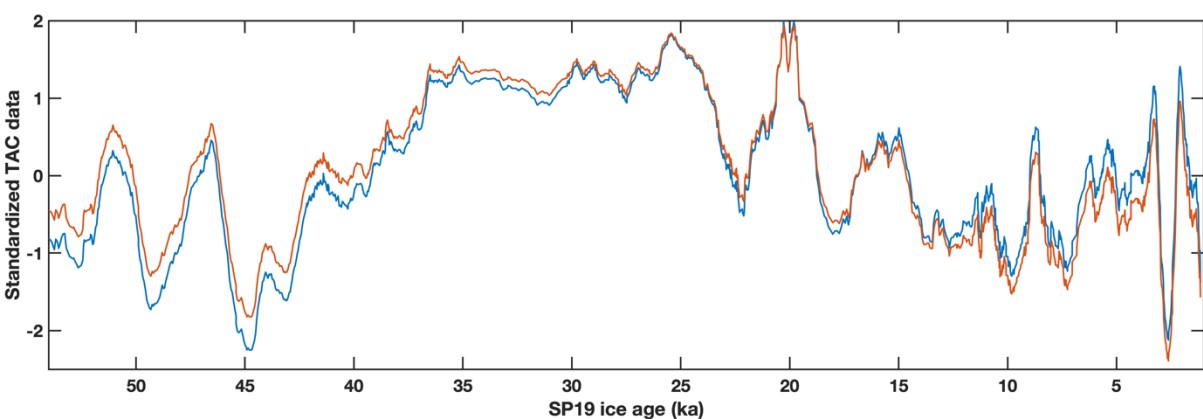


**Figure 3. Comparison of TAC and the non-thermal residual.** The non-thermal residual quantity, explained in section 3.1, explains how much temperature affects TAC. The TAC and non-thermal residual are nearly identical, indicating a very small direct effect of temperature on TAC. Blue: A standardized version of the SPC14 TAC record. Orange: Following Raynaud et al. (2007), the standardized TAC data after correcting for the direct effects of temperature ($V_{cr}$) (Martiniere et al., 1992). Both

records were smoothed using a 10-point moving average, to better highlight the difference.







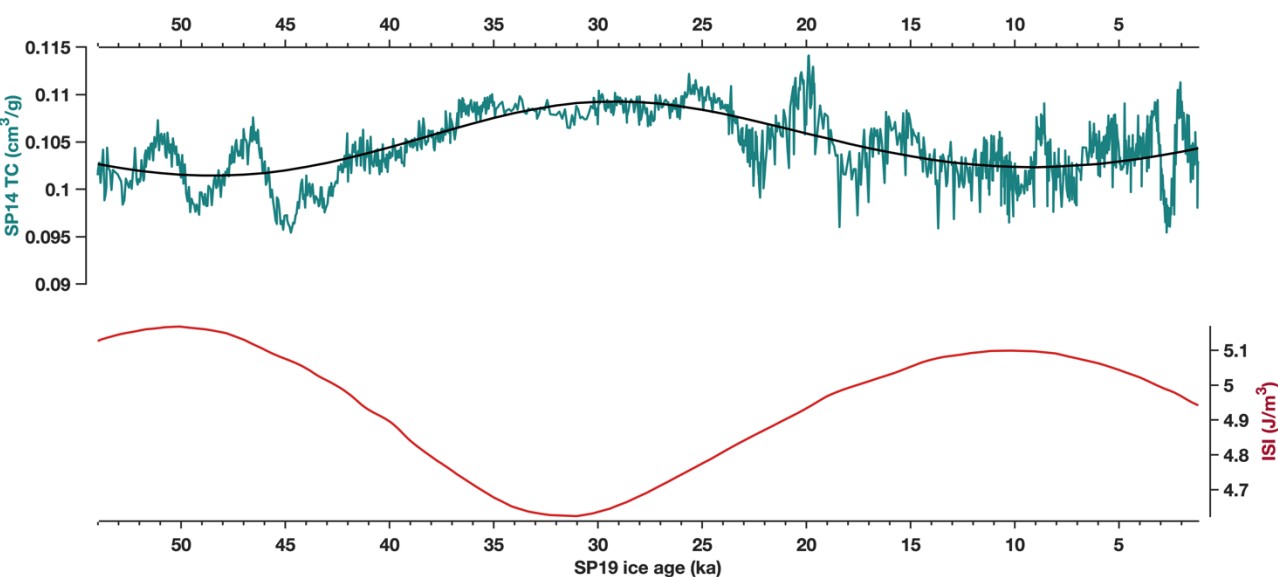

**Figure 4: SPC14 TAC predicted from integrated solar insolation (ISI)** Top: SPC14 TAC (green) and by linear regression of TAC with ISI using a cutoff threshold of 225 W/m². $R^2 = 0.46$, $p < 0.0001$, $b = 0.194$, $m = -0.0173$. Bottom: ISI at 90°S using a cutoff threshold of 225 W/m².





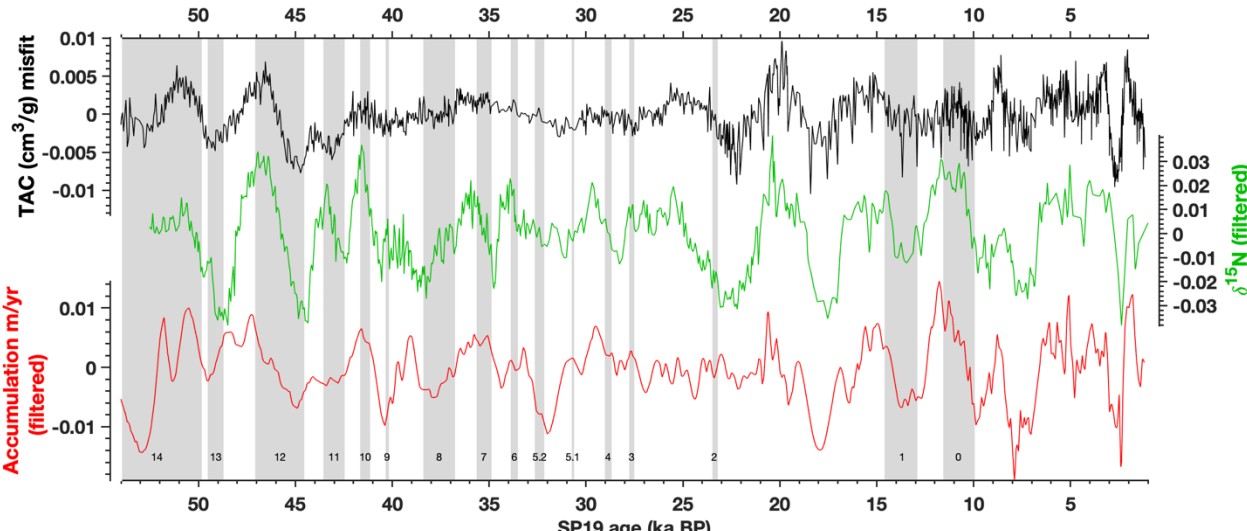

**Figure 5: TAC, accumulation rate, and δ¹⁵N-N₂.** TAC misfit (black, upper) compared with $\delta^{15}$N-N$_2$ (green, middle: Winski et al., (2019) accumulation rate (red, bottom: Kahle et al., 2021). Both the $\delta^{15}$N-N$_2$ data and accumulation rate and were filtered using a highpass filter removing variability with period greater than 10,000 years prior to comparison. TAC misfit was plotted on the SP19 ice age scale. The highpass filtered $\delta^{15}$N-N$_2$ is plotted on the SP19 gas age scale (Middle: Epifanio et al., 2019). The highpass filtered accumulation rate on the SP19 ice age scale (Bottom: Winski et al., 2019) Grey shaded areas are D-O events, numbered on the bottom for reference. Correlation coefficients are presented in Table 4.



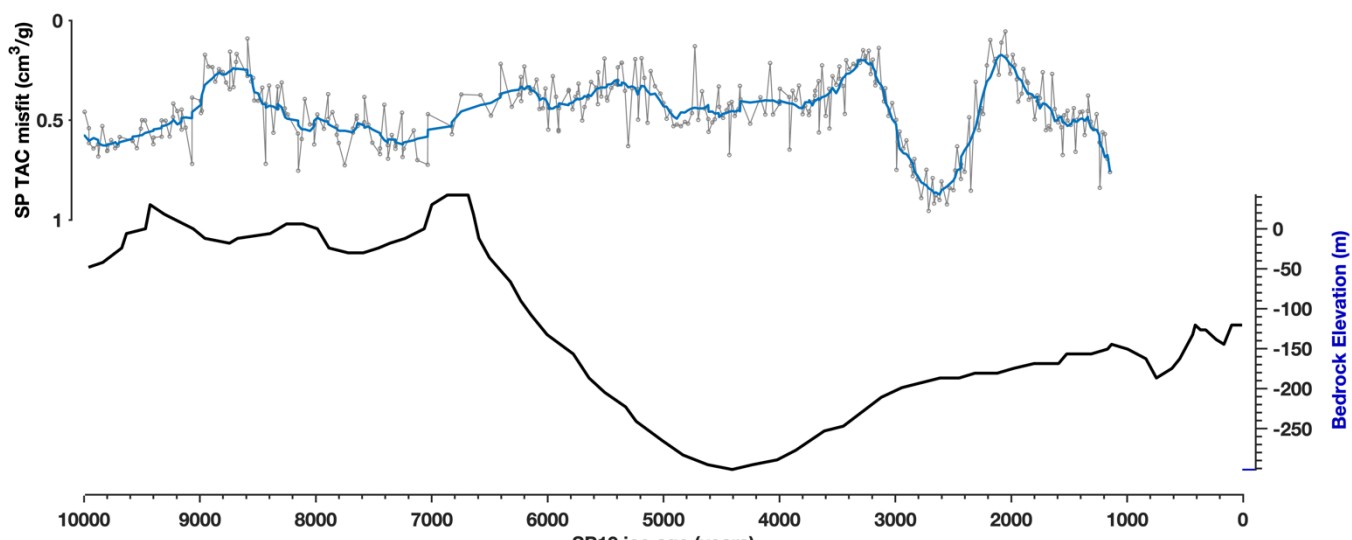

**Figure 6: SPC14 ISI regression misfit and bedrock elevation** Top: SPC14 ISI regression misfit (TAC with orbital-scale trends from ISI removed) in the Holocene. Bottom: Bedrock elevation, plotted against SPC14 ice parcel ages. Data from Kahle et al., 2021. Ice parcel ages are inferred from modeled ice parcel paths adapted from (Kahle et al., 2021). Millennial-scale changes in TAC through the Holocene have no corresponding bedrock elevation change. Zero meters is the depth of the current

bedrock at the South Pole, where the ice sheet elevation is about 2,700 m thick.




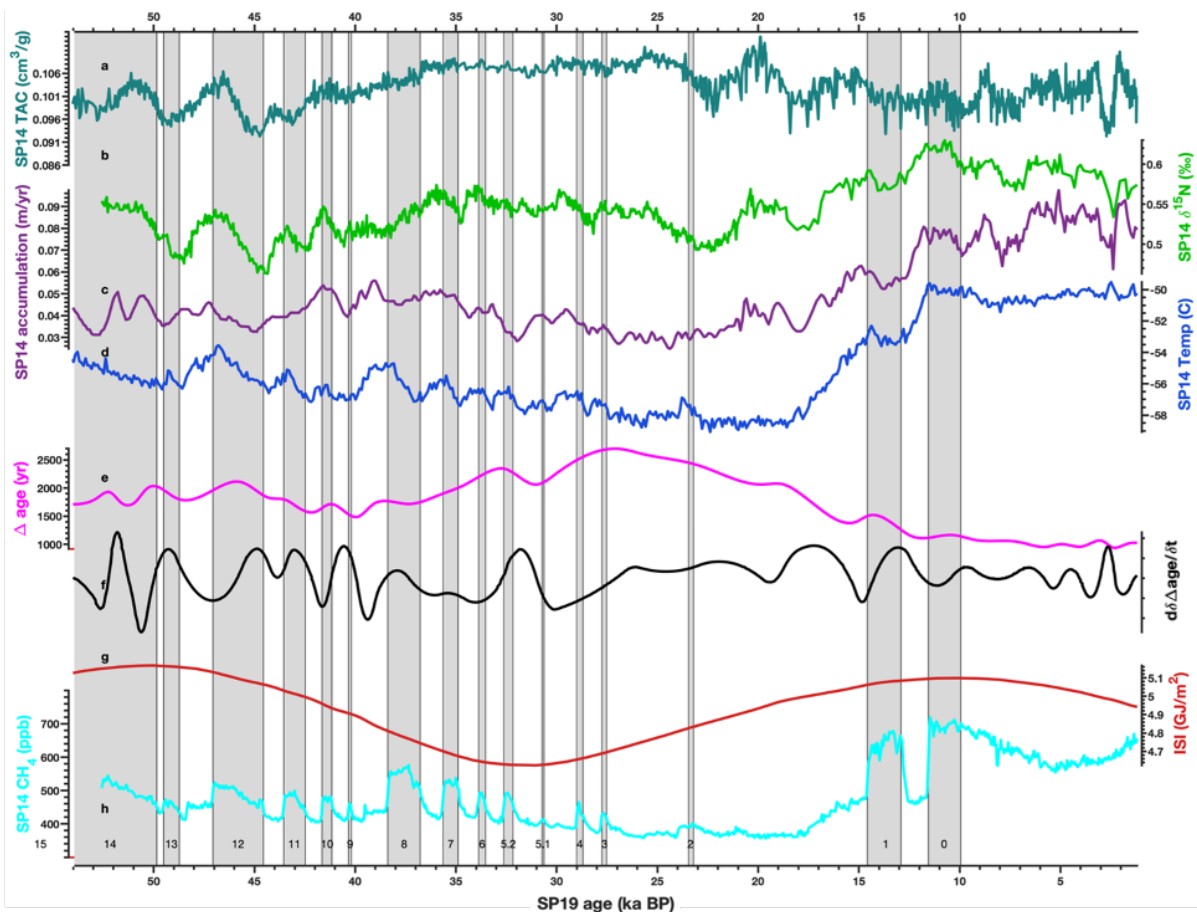

**Figure 7: TAC record of SPC14 ice core and other climate proxies**. a) SPC14 TAC record. Mean value of duplicate measurements, plotted on the ice age scale. b) $\delta^{15}$N-N$_2$, plotted on the gas age scale (Winski et al., 2019). c) Accumulation rate plotted on the ice age scale. For the last 11.3 ka the accumulation rate was constrained by layer counting of seasonal variations of magnesium and calcium ions (Winski et al., 2019). For older time periods, accumulation is modeled using an inverse method (Winski et al., 2019, Kahle et al., 2021). D) $\delta^{18}$O$_{ice}$ a local temperature proxy (Kahle et al., 2021). e) SPC14 $\Delta$age, empirically derived using independent gas and ice age markers (Epifanio et al., 2020) f) Derivative of $\Delta$age. g) Integrated Summer Insolation (ISI), at a cutoff threshold of 225 W/m$^2$, at \$90°S (Huybers et al., 2006). h) SPC14 CH$_4$, plotted on the gas age scale, added for chronological orientation (Epifanio et al., 2020). Grey bars are numbered and represent D-O events for reference.



**Figure 8: Multiple regressions of TAC compared to SPC14 TAC record.** a) SPC14 TAC data (grey), modeled reconstruction multiple regression which used modeled reconstructions of temperature and accumulation rate ($r^2 = 0.72$, $p < 0.001$) (blue), measured data multiple regression which includes $\delta^{15}N\text{-}N_2$ and $\delta^{18}O_{ice}$ ($r^2 = 0.69$, $p < 0.001$) (orange). b) Multiple regression comparison including $\delta\Delta age/\delta t$. SPC14 TAC data (grey), multiple regression using modeled reconstructions of temperature and accumulation rate, including $\delta\Delta age/\delta t$ ($r^2 = 0.76$, $p < 0.001$) (blue), multiple regression using measured parameters including $\delta\Delta age/\delta t$ ($r^2 = 0.77$, $p < 0.001$) (orange).



**Table 1: Volumes of CH$_4$ extraction line.** Volumes of CH$_4$ extraction line. Volumes listed correspond to highlighted areas in figure 2.

| Line | Volume (cc) |
|---|---|
| Removable Known Volume | 22.14 |
| Front Array | 28.45 |
| Total | 65.03 |
| GC to isolation space | 16.90 |
| Isolation Space | 6.19 |

**Table 2: Multiple regressions of TAC** Left: terms of multiple regression of TAC, when using modeled reconstructions for temperature and accumulation rate ($r^2 = 0.72$). Parameter listed is the term considered in the regression. Parameters are listed in the order of how much adjusted $r^2$ changes when the parameter is removed from to the regression. Interaction terms are not listed. All factors listed are statistically significant (null hypothesis, $p<0.05$). Right: terms of multiple regression of TAC when using measured parameters ($r^2 = 0.69$). Parameter listed is the term considered in the regression. Parameters are listed in the order of how much adjusted $r^2$ changed with their addition to the regression. Interaction terms are not listed. All factors listed are statistically significant (null hypothesis, $p<0.05$).

| Modeled reconstruction multiple regression | | Measured parameter multiple regression | |
|---|---|---|---|
| Parameter | % Change in adjusted $r^2$ | Parameter | % Change in adjusted $r^2$ |
| ISI | 15 | ISI | 10 |
| Accumulation | 14 | $\delta^{15}$N-N$_2$ | 8 |
| $\Delta$age | 8 | $\delta^{18}$O$_{ice}$ | 5 |
| Temperature | 3 | $\Delta$age | 3 |





**Table 3: Pearson correlation coefficients of single regressions.** Pearson correlation coefficients of single regressions. Highpass filtered records were filtered using a bandpass filter than eliminated frequencies greater than 10,000 years.

|  | TAC | Highpass filtered TAC |
|---|---|---|
| $\delta^{15}$N-N$_2$ | 0.13 | 0.25 |
| $\delta^{15}$N-N$_2$ (highpass) | 0.25 | 0.51 |
| Accumulation | -0.20 | 0.18 |
| Accumulation (highpass) | 0.18 | 0.59 |
| $\delta^{18}$O$_{ice}$ | -0.39 | 0.10 |
| $\delta^{18}$O$_{ice}$ (highpass) | 0.11 | -0.13 |
| Temperature | -0.37 | 0.11 |
| Temperature (highpass) | 0.11 | 0.05 |

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
