# Peer review of "Millennial and orbital-scale variability in a 54,000-year record of total air content from the South Pole ice core."

_EGUsphere, 2023_

## Author Comment (AC1)

We want to thank Reviewer #1 for their thoughtful and helpful comments, which have improved the manuscript. We greatly appreciate your input. Our response is below.

**The total air content of the SPC14 ice core is displayed here at very high resolution over the last 54,000 years. It shows orbital (albeit on a short period) and variability and variability at millennial scale on some periods. By comparing this TAC record to records of other proxies and integrated insolation curves, the authors elaborate on the mechanisms which can explain the observed variations. Accumulation rate seems to be an important control on the variations but it also seems that the mechanism at play is not the same as in Greenland.**

**In general, the manuscript is well-written and well-illustrated. I recommend its publication after the following comments are taken into account.**

**105 : Replace « = » by « is »**

Line 105, was replaced as recommended.

**153 : Can you explain how you estimate accurately the line temperature for the portion outside of the GC oven ?**

Replaced [line 153] with: *"This relied on an accurate estimation of the line temperature for the portion outside of the GC oven."* With *"The line temperature for the portion outside the GC oven was estimated by averaging the measured temperature at multiple points adjacent to the line. The variation in temperature of the line was less than 0.1°C."*

**154 : remove « is »**

Removed, as recommended

**165 : the use of V1 and V2 are confusing since V1 was used and defined before (eq. 1, l. 105) and I am not sure that it refers to the same volume. Or is it the same volume ? Please clarify.**

Thanks for pointing out this potential confusion. To avoid this, we removed subscripts on equation 1, (line 105). Also changed the paragraph that begins on [line 110] to: *"The effective temperature of the line can change from day to day due to differences in air temperature, ethanol bath temperature, chiller efficiency, and flask headspace variation due to sample size. Because of the daily variation, it is problematic to use a weighted average calculation of $T_{eff}$ in our calculation of TAC, as this would require a daily estimate of $T_{eff}$ or an assumption that it is constant. To avoid this issue, we take advantage of the methods for methane measurement. During the methane measurement, we expand the air released from the melted ice four times, with each subsequent expansion releasing lower pressures from the flask headspace. The ratio between the subsequent expansion pressures allows us to calibrate the TAC measurement, as follows."*

**Similarly, I am not sure that you refer to the same P1 and P2 than before (in equation 3, P1 and P2 were the pressures of first and second expansion).**

See the explanation, addressing Line 165, above.

**175 – can you explain clearly what is the ratio of pressures ? Which pressures ?**

  This refers to the ratio of the pressure for the second expansion relative to the first, the third to the second and the fourth to the third. Changed the last paragraph of section 2.1 (beginning line 176) to read *"Finally, multiple expansions of air from the front array to the GC volume consistently give a ratio of pressures (between the four subsequent expansions for one measurement) of 0.56. When this ratio is combined with equation (5), yields a final $\frac{T_{gc}}{V_{gc}}$ of 12.79 (K/cc)." This change, combined with the change on [line110], outlined above, give a clearer explanation of what is meant by 'the ratio of pressures'.*

**The confusions between notations noted above make it very difficult to properly understand the description of the analytical device and the way TAC is calculated. This part should be thoroughly rewritten.**

See changes detailed above.

**195 and after – not enough information is given on the cut-bubble correction for this study. Can you explain in more details how the correction has been derived and how the micro-CT measurements have been used? Is it possible to show the correction for the top 200 m since it appears that this correction is variable from sample to sample ?**

The micro-CT measurements were done by Fegyveresei et al (2018) whose methods are not yet published. The micro-CT measurements were done in conjunction with traditional bubble size and count methods, and only serve to validate the traditional methods. We propose the following edits to lines 195 and after, and the inclusion of the below figure, which outline the cut bubble correction applied.

  *"This cut-bubble correction is based on a statistical relationship between the total number and average size of bubbles in a given sample, and the amount of exposed surface area that is cut during sample preparation (Saltykov, 1976). Bubble numbers and average sizes were determined during number-density, physical property measurements, and micro-CT measurements as described in Fegyveresi et al. (2011), Fitzpatrick et al., (2014), and Fegyveresi et al. (2018). Following the methods in Martinerie et al., (1990) and Fegyveresi (2015), we interpolated the bubble size and density data to the depth of each TAC sample, and applied the correction based on the bubble size distribution for each sample across the dataset. To estimate the exposed surface area of each sample, we used standard dimensions (2.5cm x 2.5 cm x 9 cm) across all samples. There is likely some missed variation due to trimming on the edges of samples, but the variation is small." We calculated a cut-bubble correction that gives a maximum of 8% loss in the first 200 m of ice, decreasing to 1.9% TAC loss at the base of the bubbly ice at ~1200m at the onset of the clathrate-ice transition, as shown in figure 2. We applied no correction below the base of the bubbly ice."*

[Figure]

*Figure 1: Cut-bubble correction for the SPC14 ice core*

**206 : Did you try to have the TAC also on a gas age?**

Yes, we also explored plotting the TAC on the gas-age scale. Ice-age was chosen due to the proposed mechanism (both on orbital and millennial time scales) acting at the surface of the firn, which would be closer to the ice-age than the gas-age.

**Also, as you compare it later with the d15N of N2s, I imagine that d15N of N2 is on gas age and TAC on ice scale – what is the mechanistic link between the two if they are not on the same age scale?**

We compare TAC on the ice age scale with d15N-N2 on the gas age scale. The mechanistic link between TAC and d15N-N2 is through accumulation (an ice-age process). d15N-N2 is being trapped at bubble close-off, making the parameter a gas-age scale process. Our hypothesis is that d15N-N2 is reflecting the effect of firn thickness (a surface change) at the bubble close-off. Because of this delayed effect in 15N reflecting the firn thickness, we feel confident in our choice to compare d15N on the gas age scale with TAC on the ice age scale.

We suggest adding the following revision to the third paragraph of 3.4.1 after sentence 4 to incorporate this explanation:

*"...At this site, greater accumulation rates cause a thicker firn column and a subsequently higher $\delta^{15}N$-$N_2$. Because $\delta^{15}N$-$N_2$ is not set until pore close-off, toward the base of the firn, comparison of the effect in $\delta^{15}N$-$N_2$ with accumulation rate is done using the gas age scale for $\delta^{15}N$-N2 and ice age scale for TAC. Winski et al., (2019) also notes the close resemblance of $\delta^{15}N$-$N_2$ and the Holocene accumulation rate reconstruction, which is further evidence to support the use of $\delta^{15}N$-$N_2$ as an indicator of accumulation rate changes in SPC14."*

**241 : change the « x » symbol**

Changed to '*' to be consistent with the rest of the paper.

**251 : explain what are standardized versions of « TAC" and « Vcr* »**

Added to the sentence after line 252.

*"The standardized data sets were created by subtracting the mean value (of TAC or Vcr, respectively) and then dividing by the respective standard deviation."*

**290 : I know that it is explained in other places in the manuscript but it is important to document here the speed of the change. In particular, it is important to document the speed of the change because you mention that it is « abrupt ».**

Suggest changing the text on line 290 to read:

*"The approximate magnitude of the largest, abrupt. millennial-scale changes is 0.007 cm³/g in ~ 3kyr, which is similar to the abrupt millennial-scale variations observed in NGRIP, which were typically around 0.01 cm³/g in the same time frame (Eicher et al., 2016)."*

**345 : why do you mention only the resemblance between TAC and accumulation rate and between TAC and d15N of N2 ?**

- **First, you should explain on which timescale the different records are compared (the sentence « « are also highly correlated with d15N-N2 at all depth » is quite confusing – indeed, if TAC and d15N-N2 are correlated on a depth timescale, then I do not understand why TAC should be on an ice scale since d15N-N2 is on a gas timescale)**

   To avoid confusion about age scales, changed line 356 to *"Second, the millennial-scale changes in TAC, are also highly correlated with $\delta^{15}$N-N$_2$ plotted on the SP19 gas age scale ($r^2 = 0.51$, $p < 0.001$, Figure 5 and Table 2)."*

- **Second, why don't you also mention the resemblance between TAC and d18O of ice? I imagine that there is also a good correlation? What would be the r2 for the correlation between TAC and d18Oice**

   The Pearson correlation coefficient between d18Oice and TAC is quite low (-0.13), and is listed in Table 3. We also compared the temperature reconstruction of Kahle et al (2021) and also found a low r value. These results were not discussed in the text but are available in table 3.

   We suggest adding the following line at the end of paragraph 3, section 3.4.1 to alert the reader to the single regressions done on other climate parameters *"TAC was compared with temperature as well as d18Oice. Low r-values were recorded, and the results are listed in Table 2"*

- **Any link between millennial variations of TAC and millennial variations of dust concentration? What would be the r2? Dust load can indeed also influences grain size and this influence has not been discussed in this manuscript. It is important to add a few sentences on this possible influence in a revised manuscript.**

The correlation between dust and TAC is very low, r2 = 0.03. Dust levels at the South Pole are also very low, so we theorize that dust would not likely have a large impact at this site. We suggest revising the last paragraph of section 3.4 to reflect this.

*"Other hypotheses for changing TAC include layering due to melt, and dust affecting grain metamorphism. Layering due to melt or other effects influences the trapping of air in ice, shaping TAC. However, due to the lack of melt layers at this location, this possibility is beyond the scope of this study to investigate. Dust has also been documented to influence grain metamorphism in the firn. Due to its interior location, the ice at South Pole experiences very small dust flux. We observe no correlation between dust deposition and TAC."*

- **It is really interesting that the TAC signal at SP can not be explained the same way as the TAC signal at NGRIP. However, it would be great to elaborate a bit more and provide one figure showing the comparison between the two records and their relationship with accumulation rate so that the reader understands clearly the different relationships between TAC and accumulation rate in the two sites.**

  Eicher et al (2016) used stacked data from D-O events (defined by multiple parameters including temperature, d18O, CH4, and d15N) and stacked TAC data to show a delayed change in TAC due to rapid climate changes. However, as per the reviewers request, we suggest adding the below figure and caption after figure 5 which shows the *comparison of TAC and accumulation at NGRIP:*

[Figure]

***Figure 6: TAC and accumulation rate at North GRIP.*** *TAC (grey, upper: Eicher et al., 2016) compared with accumulation rate (red, bottom: Kindler et al., 2014) Black line is smoothed TAC using a 10-point running average. Grey shaded areas are D-O events, numbered on the bottom for reference. Correlation coefficients are presented in Table 4.*

In addition, we suggest changing the first sentence of the last paragraph of section 3.4.1 to read: *"Studies of TAC in Greenland suggest a different mechanism for similar-magnitude changes in TAC. Eicher et al. (2016) observed a complex, asynchronous relationship between rapid climate changes (D-O events) and millennial-scale TAC changes in the NGRIP ice core. Figure 6 shows the Greenland (North GRIP) TAC record compared with accumulation at the same site."*

- **I am not sure to support the first sentence of section 3.5. Indeed, if the dependence of TAC on accumulation rate (or other influences) is not the same on different sites (+ this study does not provide a clear mechanism), we should be very cautious in using the finding on SP to better interpret « future TAC record » since the controls may be different.**

We agree that this multiple regression is not meant to be a 'solution' for how TAC responds to multiple parameters at all ice core sites. We suggest the following revision to the first sentence of section 3.5:

*"A multiple-regression analysis was performed to examine how climate-related variables correlate with TAC at SPC14. This analysis was performed to examine the possibility of removing non-elevation-dependent signals from the record."*

- **The multiple regression is a bit difficult to follow. Indeed, while we can assume that ISI and accumulation rate are largely independent, there is strong links between d18Oice, d15N-N2, Dage and accumulation rate so that I do not really understand why the multiple regression is not simply done on ISI and accumulation rate (or ISI and d15N-N2) ? The choice of the multiple regression on 4 parameters, 3 of them being strongly linked should be much better explained.**

The multiple regression is done on a variety of climate parameters to create a regression that fits the data best. D15N, d18Oice and Dage are all correlated through complicated climate relationships. However, their inclusion in the multiple regression serves to increase the goodness of fit of the regression model. A model that uses only accumulation and ISI (modeled parameter multiple regression) or d15N and ISI (measured parameter multiple regression) produces an adjusted $R^2$ value of 0.51, and 0.62, respectively, which are still very strong goodness of fit values. Adding the other climate parameters only enhances the adjusted $R^2$ value, and in the case of dDage it also helps to explain the reason behind why the predictive power of the model increases (see explanation below).

If requested, we could replace section 3.5 with:

*A multiple-regression analysis was performed to examine how climate-related variables correlate with TAC at SPC14. This analysis was performed to examine the possibility of removing non-elevation-dependent signals from the record. Because we do not expect large elevation changes at the South Pole site, SPC14 is an excellent ice core to examine this possibility. If the TAC variability in the SP14 core can be explained using measured or modeled*

*climate variables, it might be possible in future projects to extract the portion of the variability due to elevation change. Here we considered two separate multiple linear regression analyses. In the first multiple regression (referred to as the 'modeled reconstruction' multiple regression), we considered ISI and accumulation rate. In the second multiple regression (referred to as the 'measured data' multiple regression), we considered ISI and $\delta^{15}N$-$N_2$. TAC data and variables considered are plotted in Figure 7.*

*The modeled reconstruction regression included ISI and the Kahle et al. (2021) reconstruction of accumulation rate. The modeled reconstruction multiple regression had a maximum adjusted $r^2 = 0.51$ ($p < 0.0001$), therefore the combined relationship accounts for 51% of the variation in the SPC14 TAC. The modeled reconstruction multiple regression residuals show an even distribution. The parameters are listed in Table 2 in order of how much each parameter affected the adjusted correlation coefficient.*

*A regression using only measured parameters incorporated $\delta^{15}N$-$N_2$ instead of using the modeled accumulation rate. Results for the measured data multiple regression are listed in Table 3, again in the order of how much each variable changes the final multiple regression's adjusted correlation coefficient. We find a maximum adjusted $r^2 = 0.62$ ($p < 0.0001$). Both the modeled and measured parameter multiple regressions compare well (Figure 8).*

*For both the modeled and measured regression models, addition of other climate variables increased the goodness of fit. Adding temperature and $\Delta age$ to the modeled multiple regression increased the $r^2$ to 0.72. Adding $\delta^{15}N$-$N_2$ and $\delta^{18}O_{ice}$ to the measured parameter multiple regression increased the $r^2$ to 0.69. While adding these parameters increased the goodness of fit of the models, suggesting that they do record phenomena important to controlling TAC, the other climate parameters are also highly correlated between themselves, which makes the interpretation of the regression parameters difficult (Gregorich et al., 2021)."*

Add reference: Gregorich M, Strohmaier S, Dunkler D, Heinze G. Regression with Highly Correlated Predictors: Variable Omission Is Not the Solution. Int J Environ Res Public Health. 2021 Apr 17;18(8):4259. doi: 10.3390/ijerph18084259. PMID: 33920501; PMCID: PMC8073086.

- **ISI and accumulation account for 14 and 15% of the multiple regression (l. 422). This is quite weak. Would these proportions be larger if the multiple regression is done only on ISI and accumulation rate?**

  The proportions of how much each parameter adds to the multiple regress would increase if fewer variables were used. Using only ISI and accumulation rate to create a multiple regression, the absence of ISI would decrease the regression r2 by 0.25, and removing the accumulation term would decrease the regression r2 by 0.15.

- **The influence of the dDage/dt is discussed but does not help to identify the mechanism at play (l. 439 : « the reason dDage/dt helps explain TAC changes in the firn is not at first clear ») so why not exploring the influence of dAccu/dt or d(d15N-N2)/dt or … ? The choice of the parameters used in the multiple regression line should be much more discussed.**

We recommend changing paragraph 4 of section 3.5, explaining the analysis of dDage/dt to read:

*"Large misfits between the multiple regression solution and measured TAC seem to occur during times when the climate is rapidly changing. An interesting feature of this analysis is that if the derivative of Δage (dΔage/dt) is added to the multiple regression, it seems to explain more of the variability observed in the TAC record. A comparison between a regression that includes dΔage/dt, and a regression that does not include dΔage/dt is shown in Figure 8. Specifically, dΔage/dt seems to correlate well with the magnitude of TAC change that occurs at 2,600 years as well as the large variations that occur between 45 ka and the oldest part of the record. A regression analysis that includes $\partial\Delta age/\partial t$ and the measured parameters (ISI, $\delta^{15}N$-$N_2$, $\delta^{18}O_{ice}$, and Δage) gives an adjusted $r^2$ of 0.77 (p < 0.0001), meaning that dΔage/dt and its interactions describe about 8% of the measured data multiple regression solution. Adding dΔage/dt to the modeled reconstruction multiple regression increases the $r^2$ adjusted by 4%.*

*A possible explanation for why dΔage/dt explains this extra variation is that Δage responds to changing climate conditions, and times when Δage is changing rapidly (large dΔage/dt) correspond with large changes in temperature and accumulation rate. We specifically observe this at D-O 12 and 13. This agreement between large dΔage/dt and rapid climate changes again points to a mechanism in the firn column that responds to transient accumulation changes. Following the reasoning of Eicher et al (2016), times of large changes in accumulation may not allow the firn to form spherical bubbles, creating less space, and therefore lower TAC values."*

- **445 : The influence of ISI on TAC is not so obvious because the record is short. Is it possible that the effect of accumulation rate on TAC is inhibited because the ISI is on a minimum and thus inhibits the metamorphism mechanism leading to grain size modification?**

This is a possible explanation for the lack of variation in the TAC from ~25 to 35ka.

We suggest adding at the end of section 3.5 *"A possible explanation for the lack of variation at that time could be that the effect of accumulation rate on TAC is inhibited when ISI is at a minimum. This inhibition of accumulation effects on grain size could be due to ISI dominating the grain metamorphism mechanism during that period. However, more detailed studies including high resolution TAC through multiple orbital cycles would be needed to address this question."*

- **The conclusion starting on l. 467 is surprising: why isn't the influence of accumulation rate on Dage and d15N-N2 not mentionned ? How much can the influence of accumulation rate on both TAC and d15N-N2 (Dage) explain the strong link between TAC and d15N-N2 (Dage) ? I feel that some explanations are missing here so as not to give the impression of a circular reasoning.**

Revise conclusion (2) to better explain why further high-resolution data sets of Dage and d15N-N2 are required:

*"(2) Further understanding of links between $\delta^{15}N$-$N_2$, $\Delta$age, and TAC in ice cores. Accumulation rate can influence the $\delta^{15}N$-$N_2$ and $\Delta$age depending on climate, and therefore influence the TAC differently at different sites. High-resolution sampling of $\delta^{15}N$-$N_2$ in ice cores, and model-independent $\Delta$age determinations, could be used in future work to corroborate findings from the SPC14 ice core. Additionally, TAC sampling at locations with well-known accumulation rate histories will provide further constraints."*

---

## Author Comment (AC2)

We want to thank Reviewer # 2 for their thoughtful and helpful comments, which have improved the manuscript. We greatly appreciate your input. Our response is below.

**The manuscript presents new total air content (TAC) data from south pole core SPC14. The data covers 54kyr in quite high resolution and although measured in two laboratories in consistent quality. The manuscript is well structured and well written starting out with a comprehensive introduction lining out the problems with total air content (TAC). The manuscript makes it clear that it is not solving the riddle but adding another piece to the puzzle. It is a step forward in our understanding of TAC offering some hypothesis that are however not consistent with all features seen in TAC data from the northern hemisphere. I have a few questions and suggestions below and suggest publication with minor revisions.**

**Minor comments:**

**Page 4: Given that each flask will be slightly different and the amount of ice too, the volume of the setup is changing. How are you taking this into account?**

This is considered by utilizing Tgc/Vgc instead of calculating a precise volume for each flask and sample combination. We can do this because of the consistent ratio of pressures between expansions of sample air into the GC, and the calibration for Teff that was completed and explained in section 2.1 Equation 8 demonstrates how we use Tgc/Vgc instead of air sample volume to measure TAC.

To make it clear that the method can measure TAC independent of flask and array temperatures and volumes, we recommend adding the following line to the last sentence of paragraph 1, section 2.1:

*"Updates to the methods at OSU, allow the TAC measurements to be made independent of the flask and array temperatures and volumes. The methods allowing this are described in detail below."*

**Line 201: Clathrates close to the surface are probably opening when evacuated. A correction for this effect may be appropriate. I suggest to add a statement that the correction should probably be a constant in the clathrate zone and at most 1.9%.**

We agree that the correction would be small, and no more than recorded at the base of the bubbly ice. We added the following statement to the bottom of the last paragraph of 2.2 *"While clathrate ice will still have a gas-loss correction, it is likely constant and no more than 1.9%. We applied no correction after the base of the bubbly ice."*

**Line 278: see comment to figure 4**

The insolation and TAC are not shifted. We found a strong anti-correlation between the two parameters, which is in-line with previous findings, and strengthens the conclusion that ISI can be used for orbital tuning of ice cores (Raynaud 2007, Eicher et al., 2016).

**Line 320-340: Hard to follow and some repetitions, please revise this section. A more straight forward argumentation seem to me to plot the ice sheet elevation from where the ice originates versus age. –**

Deleted duplicate sentences, which reference figure 6.

We suggest, also due to comments from reviewer 3, that we delete figure 6. The comparison of bedrock elevation to TAC is hard to follow, and upstream elevation is previously published (Lilien 2018, Fudge 2020). Figure 6 detracts from the explanation, and the narrative of ice sheet elevation over time is a better explanation than the addition of a figure.

**Line 363: delete "when"**

Deleted.

**Line 365-376: I seem to understand that low accumulation leads to denser firn therefore lower TAC. What about d15N?**

This relationship is explained in paragraph before lines 365-376. Changed the last two sentences of the previous paragraph to clarify. D15N does not act on the firn column but is a secondary indicator of accumulation. Its correlation to TAC is further evidence of the accumulation effect on TAC.

"*As temperature variations are relatively minor at the South Pole, accumulation variation drives the observed changes in SPC14 $\delta^{15}N$-$N_2$. At this site, greater accumulation rates cause a thicker firn column and a subsequently higher $\delta^{15}N$-$N_2$. Winski et al., (2019) notes the close resemblance of $\delta^{15}N$-$N_2$ and the Holocene accumulation rate reconstruction, which is further evidence to support the use of $\delta^{15}N$-$N_2$ as an indicator of accumulation rate changes in SPC14.*'*

**Line 377-385: If the orbital and millennial effects were the same you should also see an orbital signal in d15N. Do you?**

We do not observe a strong orbital signal in d15N. The r2 between ISI and d15N is 0.06, indicating a weak orbital signal in d15N. This is probably due to the d15N signal being dominated by millennial scale features.

**Figure 1: Please add a depth scale to that graph so that the location of the bubble-clathrate transition can be identified.**

Recommend revising figure 1 to include a depth -age inset as shown below:

[Figure]

***Figure 1: Total air content of the SPC14 ice core.*** *(Left) Measurements are individually shown, plotted on the SP19 ice age scale (Winski et al., 2019). Black line is the smoothed record using a running 10-point average. TAC is expressed in units of $cm^3$ air at standard temperature and pressure, per gram of ice. Orange markers are TAC measurements collected at OSU (depths 130 – 841m, 1150-1751 m, pooled standard deviation = 0.0006 $cm^3$/g). Blue markers are TAC measurements collected at PSU (depths 130-1150 m, pooled standard deviation = 0.002 $cm^3$/g). (Right) Ice age as a function of depth. Data from Winski et al., (2019).*

**Figure 3: Should refer to section 3.2. Please explain how the standardization is constructed, although it is explained in the referenced papers. What is the purpose of the standardization?**

Changed the caption to reference 3.2.

Added to the sentence after Line 252: *"Following Raynaud et al. (2007) and Lipenkov et al. (2011), we then create standardized versions of TAC and Vcr, TAC\* and Vcr\*, in order to compare TAC and the non-thermal residual. The standardized data sets were created by subtracting the mean value (of TAC or Vcr, respectively) and then dividing by the respective standard deviation."*

**Figure 4: I don't understand how the minima and maxima from the insolation and the and from TAC from linear regression can be shifted. Please explain.**

The insolation and TAC are not shifted. We found a strong anti-correlation between the two parameters, which is in-line with previous findings, and strengthens the conclusion that ISI can be used for orbital tuning of ice cores (Raynaud 2007, Eicher et al., 2016).

---

## Author Comment (AC3)

We want to thank Reviewer # 3 for their thoughtful and helpful comments, which have improved the manuscript. We greatly appreciate your input. Our response is below.

**The manuscript contributes to deeper understanding of the complicated nature of the variability of the total air content of polar ice by providing and interpreting the high-resolution air content record from the South Pole ice core covering the last 54 ka. The local insolation effect on TAC is confirmed for the site without a diurnal cycle in solar insolation and with an accumulation rate that is 3 times higher than at the Antarctic sites where this effect was first discovered (Dome C and Vostok), thus further promoting TAC as a useful tool for orbital dating of the ice cores. The high resolution of the obtained record made it possible to study millennial-scale variations in TAC and to relate them to changes in the snow accumulation rate. This relationship appears to be different from that earlier observed in and explained for the NGRIP ice core, even though the amplitude of the millennial variations of TAC is similar in both cores. The authors propose a rather plausible mechanism by which pore volume at the close-off can be affected by changes in accumulation through accompanying changes in grain size in the near-surface snow. I would only suggest that the authors develop the description of this mechanism a little in order to make it clearer and more consistent with what has already been published on this topic. They also attempt to explain the difference in mechanisms linking TAC to accumulation at the cold and relatively dry sites in Antarctica and at the warmer sites with higher accumulation in Greenland, and this explanation also sounds quite plausible.**

**In general it is a good paper, but it needs a number of minor improvements and corrections (see my comments below).**

**-L39-40. 'For temperature, Martiniere et al. (1992) demonstrated a spatial correlation between site temperature and pore volume at close-off, using data from late Holocene ice core samples'.**
**Since this spatial correlation is mentioned for the first time in the manuscript, it is more correct here to refer to the 1979 paper by Reynaud and Lebel in which it was initially presented.**

Changed lines 39-40 to include the Reynaud and Lebel reference.  Changed to: *"For temperature, Raynaud and Lebel (1979), first introduced a spatial correlation between site temperature and pore volume at close-off. This was later refined by Martiniere et al. (1992) using data from late Holocene ice core samples."*

**- L45-48 'The proposed mechanism for this relationship requires that higher local summer insolation increases the size of snow grains in the first few meters of firn, which then decreases the pore volume in these same layers as they reach bubble close-off (Raynaud et al., 1997, Arnaud, 2000).' Replace Raynaud et al., 1997 with Raynaud et al., 2007.**

Replaced, as suggested.

**2.1 Total air content measurements.**

**-The description of the measuring technique, though it is detailed in many technical aspects, lacks important information about absolute accuracy of the TAC measurements and their reproducibility. The latter is important for evaluating the contribution of experimental uncertainties to the total variance of the experimental TAC record.**

We suggest adding the following text at the bottom of paragraph 1 of section 3.1:

>"*Samples measured in duplicate at OSU have a pooled standard deviation of 0.0006 cm³/g and samples measured at PSU (130 -1150 m) have a pooled standard deviation of 0.002 cm³/g. Differences in methods between the OSU and PSU labs created a mean offset of 0.0072 cm³/g. To correct for this offset, PSU values were increased to be comparable to OSU values. The pooled standard deviation of measurements for the combined dataset (130 – 1150 m) is 0.002 cm³/g, data from 1151 -1751 m OSU only have a pooled standard deviation of 0.0006 cm³/g. Data are available at the USAP data repository including details in standard deviation and sample resolution of datasets (Epifanio, et al., 2022).*"*

**-Caption for Fig 1: 'Measurements are averaged duplicate measurements'. Does it mean that for each depth two parallel samples were measured and the average value is shown in the figure? If so, please provide the discrepancy between the individual measurements. If not, please explain what you meant to say.**

See revised caption, below and new figure 1, in response to reviewer 3's earlier comments about differences between OSU and PSU data.

[Figure]

***Figure 1: Total air content of the SPC14 ice core.*** *(Left) Measurements are individually shown, plotted on the SP19 ice age scale (Winski et al., 2019). Black line is the smoothed record using a running 10-point average. TAC is expressed in units of cm³ air at standard temperature and pressure, per gram of ice. Orange markers are TAC measurements collected at OSU (depths 130 – 841m, 1150-1751 m, pooled standard deviation = 0.0006 cm³/g). Blue markers are TAC measurements collected at PSU (depths 130-1150 m, pooled standard deviation = 0.002 cm³/g). (Right) Ice age as a function of depth. Data from Winski et al., (2019).*

**-There is no information about the mass and shape of the ice samples used (important for estimating the cut bubble effect since this effect depends on the specific surface area of the samples).**

We used a standard surface area for each sample, though we recognize we are likely missing some variation here. The variation would be small across the 2300 samples. We added the following after paragraph 3, sentence 3:

*"To estimate the exposed surface area of each sample, we used a standard rectangular dimension (2.5cm x 2.5 cm x 9 cm), and a mass of 51.2 g across all samples. While there is likely some missed variation due to sample trimming, the variation is small."*

**L183-186. The amount of air trapped in refrozen ice (I wouldn't call it "solubility") depends, in addition to the air pressure in the flasks, on the number and size of air bubbles formed in this ice. These values can vary considerably from one experiment to another and are difficult to predict.**

We recognize that some air is trapped in the melted sample which is then refrozen prior to measuring the air content. Because of this, we conducted multiple melt-refreeze experiments to determine this correction (1.3%). These were done as outlined in Mitchell et al., (2015), and give consistent results.

**L193-202 Cut bubble correction.**

**1. One could understand from this text that cut bubble correction depends on the number and size of the bubbles. In fact (see Saltykov, 1976; Martienerie et al., 1990) this correction depends only on the size of the bubbles (or more precisely, on the bubble-size distribution) and on the specific surface area of the sample. How did you estimate the latter?**

We estimated the surface of each sample by using a standard size for each sample. We recognize we are likely missing some variation here, due to trimming the edges of a cubic prism shape to fit in the sample flask. The variation would be small across the 2300 samples. We added the following after section 2.2, paragraph 3, sentence 3:

*"To estimate the exposed surface area of each sample, we used standard dimensions (2.5cm x 2.5 cm x 9 cm) across all samples. There is likely some missed variation due to sample trimming, the variation is small."*

**2. Bubbles efficiently expand during ice storage at a relatively elevated temperature (e.g. at -20 ºC), so, with other things being equal, the correction increases with the time of storage and therefore bubble measurements should be done at the same time as TAC measurements.**

Because of the nature of the research being completed at multiple organizations, the TAC and bubble measurements were not done at the same time. We recognize that effect of bubble expansion occurs over time, though we do not believe this would invalidate the data set or largely change the correction. Some of our TAC measurements (26 samples) were measured in duplicate 2 years apart. These measurements have a standard deviation of 0.001 g/cm$^3$, which is well within the precision of our measurements.

**3. The correction for gas loss for bubble-free ice (i.e. ice containing only hydrates) is needed in the same way as it is needed for bubbly ice, because the ice sample loses its gas from cut hydrates as it does from the cut bubbles. In addition, if the temperature of storage was not low enough (say, above -40…-30 ºC) many hydrates in ice can dissociate with formation of air cavities whose size also needs to be measured.**

We have no way to quantify the gas loss due to clathrates since clathrate size and density have not been measured. However, the correction would be small, and no more than recorded at the base of the bubbly ice, since bubbles contain more air than clathrates, individually.

As per our response to reviewer 2, above, we suggest the following statement be added to the bottom of the last paragraph of 2.2:

"*While clathrate ice will still have a gas-loss correction, it is likely constant and no more than 1.9%. We applied no correction after the base of the bubbly ice.*"

**L209-211. I understand that for each depth two samples were measured with the OSU vacuum line using the method described in section 2.1. Can you estimate the repeatability of the measurements and present it in the paper?**

We suggest adding the following text at the bottom of paragraph 1 of section 3.1:

*"Samples measured at OSU have a pooled standard deviation of 0.0006 cm$^3$/g and samples measured at PSU (130 -1150 m) have a pooled standard deviation of 0.002 cm$^3$/g. Differences in methods between the OSU and PSU labs created a mean offset of 0.0072 cm$^3$/g. To correct for this offset, PSU values were increased to be comparable to OSU values. The pooled standard deviation of measurements for the combined dataset (130 – 841 m) is 0.002 cm$^3$/g, data from 1151 -1751 m have a pooled standard deviation of 0.001 cm$^3$/g. Data are available at the USAP data repository including details in standard deviation and sample resolution of datasets (Epifanio, et al., 2022). To our knowledge, this is the first ice core TAC record with this resolution and length, allowing in-depth comparison with other climate proxies at a site that is not likely to have experienced significant elevation change over the last 54 ka (Fudge et al., 2020, Lilien et al., 2018)."*

**L214-217 It is not clear from the text how the data from OSU and PSU were combined (and averaged?). Were the PSU measurements made at the same depths with the same resolution as in the OSU? Might it be useful to show the OSU and PSU data (after correction for the offset) in figure 1 with a different color? Please explain and comment.**

OSU and PSU data were not made at the same depths or in the same resolution, so the samples were not averaged between labs. Instead, the datasets were combined including data from both labs at individual points. The individual lab measurements are publicly available at the USAP data repository and listed in the references.

We suggest including the following figure and revised caption which uses different colors to show the two lab measurements after combining the datasets, and includes an age/depth scale as suggested by reviewer 2:

[Figure]

*Figure 1: Total air content of the SPC14 ice core. (Left) Measurements are individually shown, plotted on the SP19 ice age scale (Winski et al., 2019). Black line is the smoothed record using a running 10-point average. TAC is expressed in units of $cm^3$ air at standard temperature and pressure, per gram of ice. Orange markers are TAC measurements collected at OSU (depths 130 – 841m, 1150-1751 m, pooled standard deviation = 0.0006 $cm^3$/g). Blue markers are TAC measurements collected at PSU (depths 130-1150 m, pooled standard deviation = 0.002 $cm^3$/g). (Right) Ice age as a function of depth. Data from Winski et al., (2019).*

**L245-248. About the Vcr.**
**1. The temperature used in eq. 12 (Ts) and the one in the 'gas law'(Tc in eq. 11) are different temperatures. The first one refers to the time when snow was deposited at the ice sheet surface (corresponds to the age of the ice), the second one – to the time of pore closure (~gas age). Did you distinguish these temperatures when calculating Vcr using temperature reconstruction from Kahle et al (2021)? And if so, please explain how you did this, especially for the transient climatic conditions.**

We use the same temperature for both the surface and the close-off depth. In reality, the latter is a little bit higher due to geothermal heat, but the difference is negligible for the current application.

We add this explanation directly after equation 13: *"We use the same temperature for the surface (Ts) and the close-off depth (Tc). While the temperature at pore close-off (Tc) is a bit warmer than Ts due to geothermal heating, the difference is small when the firn column is in*

*equilibrium.”*

**It seems something is missing in the sentence 'For the temperature at bubble close-off (Ts), a temperature reconstruction from Kahle et al (2021)'. Also, please replace Ts with Tc in this sentence.**

We suggest changing the sentence to read:

*"For the temperature at bubble close-off (Tc), we used the temperature reconstruction from Kahle et al (2021)".*

**2. Eq. 12 shows the present-day (late Holocene) spatial relationship between pore volume at close-off and mean annual surface temperature. It is very unlikely that this relationship was the same in the past, especially during periods with different from today's insolation. So strictly speaking one cannot use eq. 12 to calculate Vcr.**
**L253-256. '…Vcr is a quantity that essentially describes TAC in the absence of temperature effects…'**
**Even if we assume that eq. 12 is valid for the past, the Vcr calculated from eq. 13 will contain a significant summer-temperature signal, because the impact of changing insolation on the Vc is transmitted through corresponding changes in summer temperature and temperature gradients that affect the snow metamorphism near the ice-sheet surface (Raynaud et al., 2007; Lipenkov et al., 2011).**

We are assuming the spatial relationship between pore volume and temperature remains the same. To our knowledge, the spatial relationship between pore volume and temperature has not been significantly updated. Additionally, the high $r^2$ value between TAC and Vcr indicate that the summer temperature signal that affects firn metamorphism at the ice sheet surface is small.

Suggest changing line 253-256 to read *"Vcr\* is a quantity that describes TAC if temperature did not affect pore volume at close-off, Vcr\* is a useful quantity to understand the magnitude of the direct effects of temperature."*

**L260-261. Correct references here: Raynaud et al., 2007; Lipenkov et al., 2011; Eicher et al., 2016.**

Corrected references.

**L317-319: I cannot understand this sentence.**

Changed to read *"Accumulation rates during the glacial period were just 3 cm/yr (water equivalent), which would mean the maximum amount of elevation gain due to accumulation alone would be only about 80 meters, without considering the effects of ice layer thinning."*

**L336-337. Please provide a reference for this hypothesis or justify it.**

After review, we suggest revising the last paragraph of section 3.4 to read as below. This removes the hypothesis of firn stretching affecting TAC, as this hypothesis is not supported by observation.

*"Other hypotheses for changing TAC include layering due to melt, and dust affecting grain metamorphism. Layering due to melt or other effects influences the trapping of air in ice, shaping TAC. However, due to the lack of melt layers at this location, this possibility is beyond the scope of this study to investigate. Dust has also been documented to influence grain metamorphism in the firn. Due to its interior location, the ice at South Pole experiences very small amounts of dust flux. We observe no correlation between dust deposition and TAC."*

**The need for Figure 6 doesn't seem obvious for me, but if you decide to keep it in the paper, it should come before Figure 5 .**

Agree with removing figure 6, and references to it.

**L367-369: 'The size of the firn grains at the surface seems to predict at which density bubble close-off occurs, with larger grained firn closing off at a higher density (Gregory et al, 2014)'.In fairness, it should be noted that the mechanism by which the porosity of firn at close-off is linked to the snow grain size at the surface was first proposed by L. Arnaud (1997). Later on his model was used to qualitatively describe a possible mechanism by which summer temperature and surface temperature gradients controlled by local insolation can influence pore volume at close-off, assuming a homogenous firn column and neglecting the sealing effect on the total amount of air trapped in ice (Raynaud et al., 2007; Lipenkov et al., 2011).**

Added reference to Arnaud, 1997. At this point in the manuscript, we do not feel that the summer insolation mechanism is relevant, as we are trying to express a different mechanism (accumulation instead of insolation) for grain size affecting TAC.

**-L373-374: 'Low accumulation rates create more homogeneous, spherically shaped grains which force more air to escape the ice core, leading to lower TAC'. Please provide a reference for this statement or justify it using your own observations.**

We suggest a reference to Gregory et al., 2014 and Eicher et al, 2016. Suggested revisions are included in the paragraph for the next reviewer comment.

**-L374-376: 'We propose that a mechanism of grain size and shape affecting pore volume leads to a positive correlation between accumulation and TAC, which we observe in the SPC14 ice core'.**
**L460-461: 'We propose that a common mechanism, grain size metamorphism in the top few meters of the firn, can explain both orbital and millennial-scale times scale of TAC variations in the SPC14 ice core'.**
**Since this proposed mechanism is considered by the authors as one of the main merits of their work (along with the obtained high-resolution TAC record), I would advise them to**

**pay a little more attention to its clear and consistent description, and correct alignment with what has already been published on this mechanism in connection with orbital variations in TAC. In the present manuscript, the entire description of this mechanism is confined to a single paragraph (L365-376) and seems neither clear nor complete.**

We suggest the following edits to the two paragraphs beginning on L365 to better tie in the proposed mechanism to that of previous work in connection with orbital variations in TAC.

*"Metamorphism of the ice in the first few meters of the firn may explain the link between accumulation rate and TAC. Lower accumulation rates allow grains to remain at or near the surface for a longer time, giving grains in the firn more time to grow while they remain at the surface (Courville et al., 2007). The size of the firn grains at the surface seems to predict the density at which bubble close-off occurs, with larger-grained firn closing off at a higher density (Arnaud, 1997; Gregory et al, 2014). Because ice density is by definition, inversely proportional to porosity, higher density bubble close-off (associated here with longer time near the surface of the ice sheet, larger grain sizes, and lower accumulation) leads to bubbles with less pore volume than firn with smaller grain sizes. Lower accumulation rates may additionally allow more time for grains to become spherically shaped before close-off, where higher accumulation rates tend to close off bubbles earlier in the densification process. Because grains tend to move toward a spherical shape with enough time, due to vapor diffusion (Eicher et al., 2016), low accumulation rates likely create more homogeneous, spherically shaped grains. These spherically shaped grains force more air to escape the ice core, leading to lower TAC. (Gregory et al., 2014) noted higher gas diffusivity at lower accumulation sites, implying that at low accumulation sites, the pores are closing off later, allowing time for more spherically shaped grains. We propose that a mechanism of grain size and shape affecting pore volume leads to a positive correlation between accumulation and TAC, which we observe in the SPC14 ice core, however the microstructure and physics behind the mechanism should be explored in future work.*

*In a sense, this proposed grain size mechanism is similar to the proposed mechanism for how ISI impacts TAC on an orbital timescale (Raynaud et al., 2007, Eicher et al., 2016). ISI is hypothesized to act on TAC by changing the grain size of the firn at the surface by influencing temperature gradients in the first few meters of firn. On orbital time-scales, higher ISI increases the near-surface firn metamorphism and grain size, and decreases pore volume at close off, resulting in the inverse relationship between TAC and ISI recorded in both hemispheres (Raynaud et al., 1997, Eicher et al., 2016). In our proposed mechanism for millennial-scale variations in TAC, lower accumulation increases near-surface firn metamorphism and grain size and decreases pore volume at close-off. In both scenarios (orbital- and millennial-scale changes), grain size is set in the first few meters of the firn, though by different mechanisms, and the impact is advected to the close-off depth. We propose that the relationship between grain size and accumulation rate is responsible for the large, millennial-scale changes in TAC found in the SPC14 ice core. This mechanism is complimentary to the orbital changes in TAC imposed by ISI and creates millennial-scale changes imposed on top of the orbital-scale changes."*

**3.5 Multiple regression**

**I don't see much point in multiple regression analysis involving non-independent variables**

**that correlate with each other. The latter could be one of the reasons why the authors obtained such a weak contribution of the ISI to the total variance of TAC.**

We suggest revising section 3.5 to include regressions only between ISI and accumulation and ISI and d15N, as discussed above, in response to reviewer 1.

**Surprisingly, the authors don't even mention the so-called 'wind effect' (Martinerie et al., 1994), which could account for a significant fraction of the non-orbital variability of the air content.**

Suggest adding the following line to the bottom of paragraph 2, section 3.4:
"*The wind effect, as described in Martinerie et al., (1994) would require large-sustained wind changes over millennia that are not supported by modeling reconstructions (Goodwin et al., 2014).*"

**Technical comments**
**In general, the manuscript requires additional proofreading, as it still contains many minor technical errors. I will cite those which I managed to notice and remember.**

**L331-333 and L334-336: two identical sentences in a row.**
Deleted duplicate sentences.
**L348, L357, L427, L560: please check and correct the table numbers you refer to here.**
Table numbers have been revised and proofread.
**L371-373: the first and the second parts of the sentence seem to be poorly connected.**
See edits to paragraphs starting on Line 365 (detailed above).
**L420: table 2 is mentioned here for the first time, while table 3 has already been mentioned above (as table 4, I suppose).**
Table numbers have been revised and proofread.

**Please check units for ISI in Fig. 4. Should it be GJ/m2?**
Yes. Corrected label to GJ/m2

**Additional reference:**
**Arnaud, L., 1997. Modélisation de la Transformation de la Neige en Glace à la Surface des Calottes Polaires; étude du Transport des gaz dans ces Milieux Poreux, PhD. Université Joseph Fourier.**
Added reference.